# Fire weather index data under historical and SSP projections in CMIP6 from 1850 to 2100

Yann Quilcaille[1*], Fulden Batibeniz[1*], Andreia F. S. Ribeiro[1], Ryan S. Padrón[1], Sonia I. Seneviratne[1]

[1]Institute for Atmospheric and Climate Science, Department of Environmental Systems Science, ETH Zurich, Zurich, Switzerland
*These authors contributed equally to this work

*Correspondence to*: Yann Quilcaille and Fulden Batibeniz (yann.quilcaille@env.ethz.ch and fulden.batibeniz@env.ethz.ch)

**Abstract.** Human-induced climate change is increasing the incidence of fire events and associated impacts on livelihood, biodiversity, and nature across the world. Understanding current and projected fire activity together with its impacts on ecosystems is crucial to evaluate future risks and take actions to prevent such devastating events. Here we focus on fire weather, as a key driver of fire activity. Fire weather products that have global homogenous distribution in time and space provide many advantages to advance fire science and evaluate future risks. Therefore, in this study we calculate and provide for the first time the Canadian Fire Weather Index (FWI) with all available simulations of the 6th phase of the Coupled Model Intercomparison Project (CMIP6). Furthermore, we expand its regional applicability by combining improvements on the original algorithm for the FWI from several packages. A sensitivity analysis of the default versus our improved version shows significant differences in final FWI. With the improved version, we calculate the FWI using average relative humidity in one case and minimum relative humidity in another case. We provide the data for both cases, while recommending the one with minimum relative humidity for studies focused on actual FWI values and the one with average relative humidity for studies requiring larger ensembles. The following four annual indicators: (i) maximum value of the FWI (*fwixx*), (ii) number of days with extreme fire weather (*fwixd*), (iii) length of the fire season (*fwils*), and (iv) seasonal average of the FWI (*fwisa*) are made available and illustrated here. We find that at a global warming level of 3°C, the mean fire weather would increase on average by at least 66% in duration and frequency, while associated 1-in-10-year events would approximately triple in duration and increase by at least 31% in intensity. Ultimately, this new fire weather dataset provides a large ensemble of simulations to understand the potential impacts of climate change spanning a range of shared socioeconomic narratives with their radiative forcing trajectories over 1850-2100 at annual and 2.5° x 2.5° resolutions. The produced full global dataset is a freely available resource at https://doi.org/10.3929/ethz-b-000583391 for fire danger studies and beyond, which highlights the need to reduce greenhouse gas emissions for reducing fire impacts.

**Short Summary.** We present a new database of four annual fire weather indicators over 1850-2100 and over all land area. Our first analysis shows that in a 3°C warmer world with respect to preindustrial times, the mean fire weather would on average

double in duration and intensity. The produced dataset is a freely available resource for fire danger studies and beyond, highlighting that the best course of action would require limiting global warming as low as possible.

## 1. Introduction

Anthropogenic climate change tends to make fires less predictable and exacerbate their impacts (Anon, 2019; Sanderson and Fisher, 2020). The IPCC AR6 report concluded that fire weather has become more widespread, longer-lasting, and more intense compared to preindustrial periods in some regions (e.g. Mediterranean) and that these changes are expected to increase with higher global warming levels (Seneviratne et al., 2021). This is generally associated with an increased occurrence of concurrent hot and dry conditions with increasing global warming (Seneviratne et al., 2021). Unfortunately, we have already suffered several damaging wildfires in recent years throughout the world (e.g., Australia, Turkey and Greece, Siberia, Sweden, Canada, USA, etc), with some of them formally attributed to human-induced climate change (Li et al., 2019; van Oldenborgh et al., 2021). Fires are not only destructive but also release carbon stored in vegetation, and thereby increase atmospheric $CO_2$ concentration (Lasslop et al., 2020). For example, released $CO_2$ emissions from 2019-2020 wildfires in Australia was at least higher than 35% of the country's annual amount (Li et al., 2021; van der Velde et al., 2021).

Fire weather is defined as those conditions conducive to the occurrence and sustaining of fires, and it is characterized by compound hot, dry and windy events. These conditions are increasing in frequency and intensity across many regions due to anthropogenic climate change (Abatzoglou et al., 2019; Ranasinghe et al., 2021; Seneviratne et al., 2021). Further increases in greenhouse gas forcing are likely to increase the occurrence of these compound conditions, favouring the occurrence of extreme events of fire emissions and burned area (Li et al., 2021; Jones et al., 2022; Ribeiro et al., 2022), more fire prone regions, and more complex fire dynamics. It can also induce preconditions that can exacerbate the impacts of fire, such as tree mortality (Stevens-Rumann et al., 2012) and fuel accumulation (Marlon et al., 2009).

In addition to fire weather, human activities affect fires in multiple ways. On one hand, agricultural expansion and landscape fragmentation caused by humans makes the vegetation less flammable and creates a decreasing trend in satellite observations (Jolly et al., 2015; Andela et al., 2017). On the other hand, human influence increases the fire risk in some regions due to negligence or arson. For example, the use of fire as a land clearing tool for agriculture and deforestation may ignite uncontrollable wildfires and burn large forested areas, particularly during compound dry and hot events in regions such as the Amazon (e.g. (Libonati et al., 2022; Ribeiro et al., 2022)). At the same time, the most disastrous wildfires occur in wildland and urban transition areas where human effect is the largest (Bowman et al., 2017). Even though these dynamics are hard to investigate in a future climate, it is necessary and requires projected land-use transition and socioeconomic scenarios.

Historical fire weather can be investigated with observations, remote sensing products or more spatially and temporally homogeneous reanalysis datasets (Vitolo et al., 2019). Some of these datasets cover decades and allow longer statistical analysis. For example, fire season duration is increasing according to recent long-term satellite observations (Jolly et al., 2015). Though, future fire activity is an enigma given the potential changes on ecosystems due to climate change and human activities. The interactions of human influence and climate change with fire dynamics are so complex that each ecosystem must be

studied in its own right. Therefore, improvements to fire indices and a better understanding of the interactions between mean

climate, climate extremes, humans and fire are required to project future fire activity and to mitigate its consequences.

Several indicators for fire weather have been proposed over the years (Table A.1). Although all of them have been developed to inform about fire risk, each one responded to different needs. Here, we focus on the FWI from the Canadian Forest Fire Danger Rating System (Van Wagner, 1987) for several reasons. First, this index represents potential fire danger rather than actual fire occurrence, only seizing how fire activity is prone according to meteorological conditions. The four major drivers

of fire weather at a global scale (temperature, precipitation, relative humidity, and wind speed) are accounted for, also through the impact of the moisture content of potential fuel on the fire intensity. The second reason is that clear relationships have been shown between the FWI and the burned area in Earth System Models (ESMs) (Bedia et al., 2015; Abatzoglou et al., 2018; Grillakis et al., 2022; Jones et al., 2022), making this index relevant for impact assessments. Finally, this index can be used at a global scale (Field et al., 2015) for fire danger predictability and warning systems (de Groot et al., 2015; Bedia et al., 2018)

or fire activity under projected climate change (Abatzoglou et al., 2018; Jain et al., 2020; Ranasinghe et al., 2021).

Here, we present a new dataset of FWI, based on climate data from the 6th phase of the Coupled Model Intercomparison Project (CMIP6) and using an improved algorithm. We build upon the work of (Abatzoglou et al., 2019) for the previous generation of CMIP models. The novelty of this work comes from (1) the expanded regional applicability thanks to improvements on the original algorithm, (2) using the latest CMIP data covering historical and shared socioeconomic pathways

(SSPs), from 1850 to 2100, and (3) providing the whole database to the users, thus enabling a large range of usages. Several packages have proposed different adjustments to the initial algorithm of the FWI. By gathering these improvements, our new algorithm allows us to compare the sensitivity of the product to these modifications. We produce an updated FWI dataset which enables analyses over longer time scales while considering climate sensitivity and internal variability. We envision this open-access dataset of fire weather as a valuable resource for scientists in the fields of climate change and risk assessment,

insurance companies, forestry agencies and more.

## 2. Data and Methods

### 2.1 Climate model data

We use all available CMIP6 simulations (Eyring et al., 2016) of climate models to create FWI data over 1850-2100 using the experiment *historical* and all SSPs (O'Neill et al., 2016; Tebaldi et al., 2021), namely *ssp119*, *ssp126*, *ssp434*, *ssp534-over*,

*ssp245*, *ssp460*, *ssp370* and *ssp585*. Using all SSP experiments allows us to represent high/low mitigation and adaptation challenges resulting in different radiative forcings by the end of 2100. We retrieve daily maximum temperature (*tasmax*), precipitation (*pr*), wind (*sfcWind*) and minimum relative humidity (*hursmin*) data from all available ensemble members of all available ESM. More precisely, the FWI is calculated only if the four variables are provided for the experiment. For scenarios, there is an additional condition: it depends not only on the availability for the experiment, but also if the corresponding

*historical* could be run. The algorithm of the FWI requires the last values of the *historical* as initialization for the scenario for continuity reasons (more details are provided in Section 2.b). The full list of runs calculated is represented in Figure 1.

The algorithm for the FWI requires daily temperature, relative humidity and wind speed at noon, and the daily accumulated precipitation (Van Wagner, 1987). Using variables at sub-daily resolution would significantly reduce the number of available runs. Instead of noon variables, daily maximum temperature and daily minimum relative humidity can be used, as done for

CMIP5 for one ensemble member per model (Abatzoglou et al., 2019).

Some applications may need to maximize the number of ensemble members per model, thus a second dataset is provided. The first dataset deduces the FWI from daily minimum relative humidity, while the second one use daily mean relative humidity (*hurs*) instead, because this variable is provided for more model runs. In Section 3.d, we provide a sensitivity analysis comparing the FWI based on daily average relative humidity against daily minimum relative humidity, while Figure A.1

summarizes the available runs with daily average relative humidity.

We highlight that using CMIP6 data comes with limitations. Although this is the result of a large community effort (Tebaldi et al., 2021), there may be some biases and discrepancies in these inputs (Wilcox and Donner, 2007; Rossow et al., 2013; Pfahl et al., 2017; McKitrick and Christy, 2020). Analysis of these biases have been performed for temperatures in (Fan et al., 2020), regional precipitations (Rivera and Arnould, 2020; Agel and Barlow, 2020; Ajibola et al., 2020), relative humidity (Douville

et al., 2022) and wind (Shen et al., 2022). A bias-corrected version of CMIP6 data may be used as inputs, but existing datasets do not provide the necessary variables for the computation of the FWI (Carvalho et al., 2021; Xu et al., 2021), nor the full ensemble that we use here.

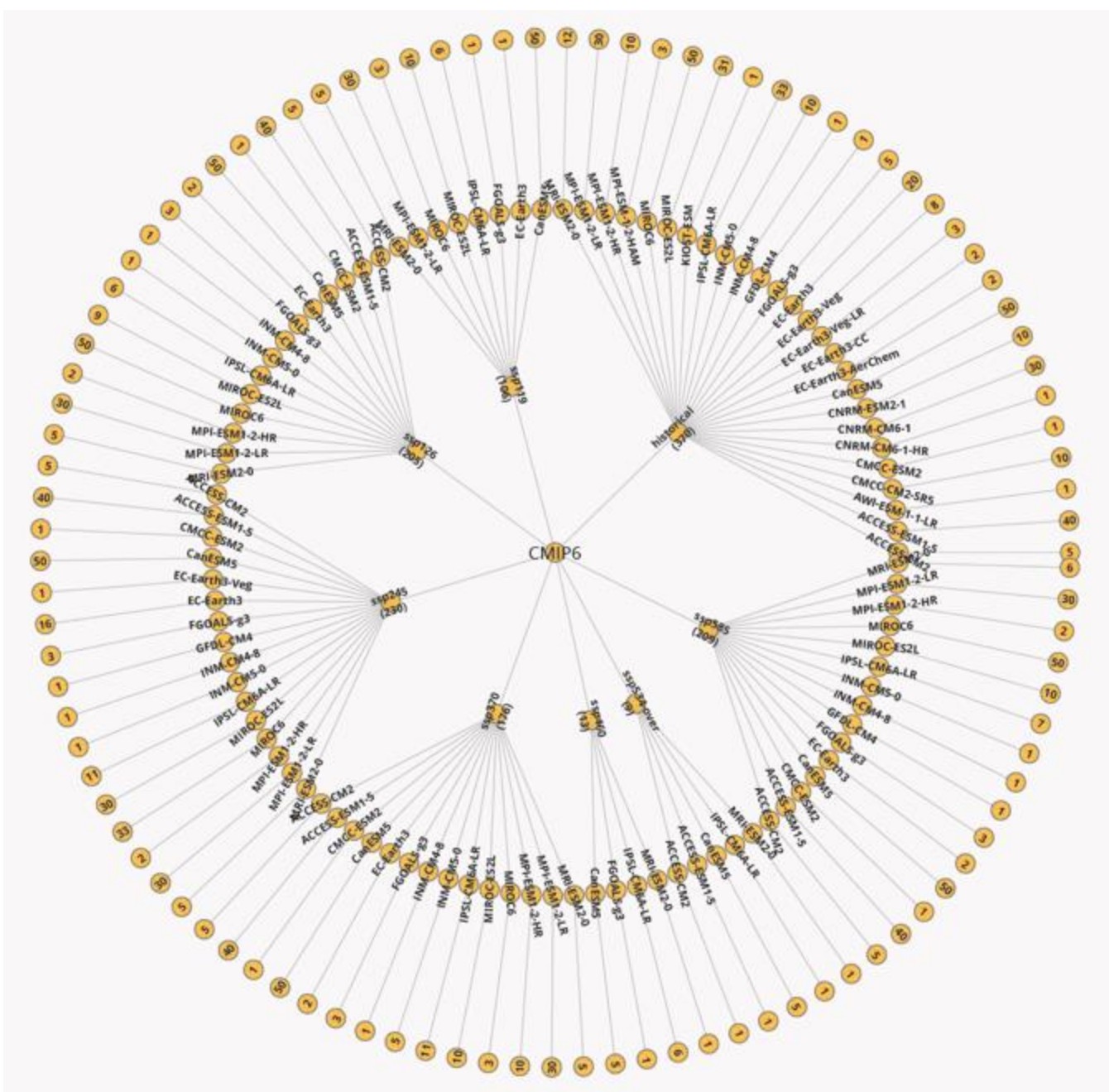

**Figure 1.** Runs used throughout the study. For each experiment, each ESM is selected if at least one ensemble member is available and valid for all the variables used as input for the calculation of the FWI. The outermost circle indicates the corresponding number of ensemble members of each ESM and scenario. Altogether, 1321 runs are used here.

## 2.2. Fire weather index and adjustments

The FWI system consists of several indices calculated in three steps (Van Wagner, 1987; Lawson and Armitage, 2008) as
illustrated in Figure 2. The moisture contents of organic materials are calculated first, through the Fine Fuel Moisture Code
(FFMC), the Duff Moisture Code (DMC) and the Drought Code (DC). The FFMC rates the moisture content of fine fuels and
of the litter, hence the probability of ignition. The DMC rates the moisture content of slightly compacted organic layers at
medium depth, giving a sense of the fuel consumption. The DC rates the moisture content of deep and compact organic layers,
depicting the behaviour of slow burning materials and representing seasonal effects. These three indexes are actually book-
keeping systems, accounting for changes in moisture through precipitation and drying. It is crucial to note that they are unitless
and that they are defined "with values rising as moisture content decreases for the best psychological effect" (Van Wagner,
1987), as illustrated with equation (5) of (Lawson and Armitage, 2008).

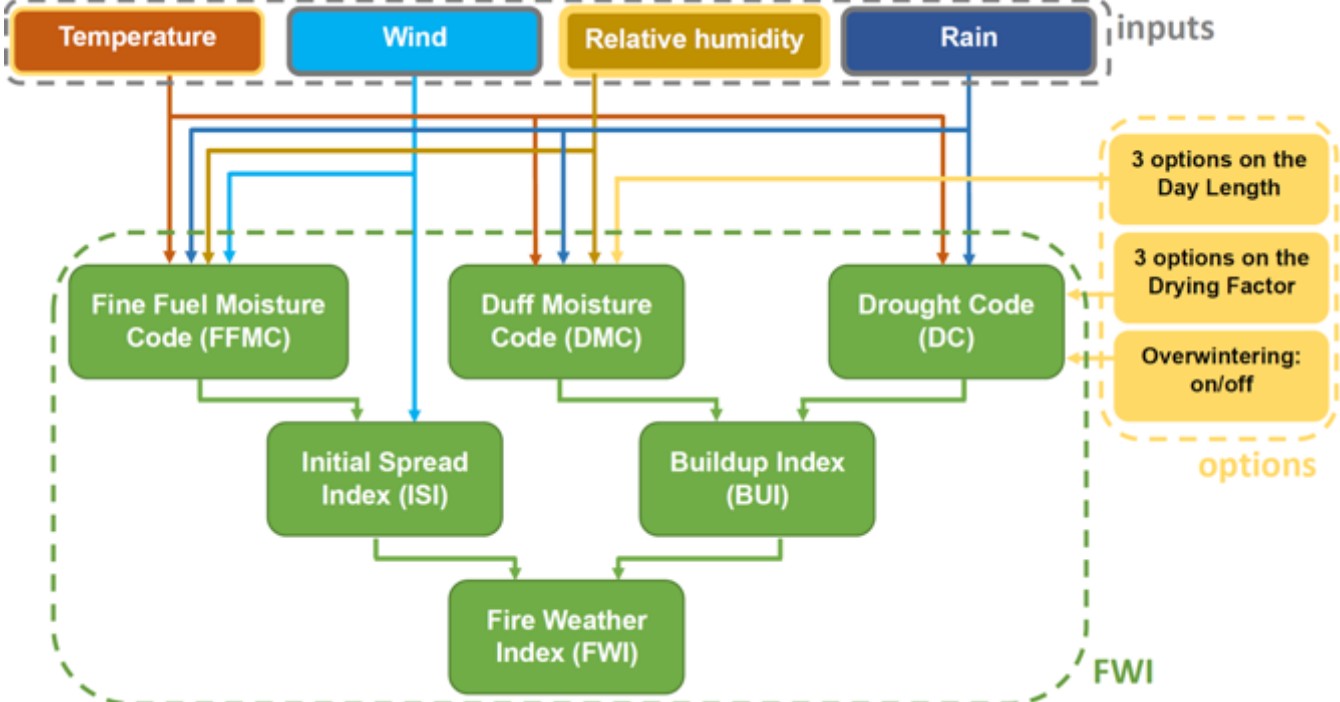

**Figure 2.** Conceptual structure for the calculation of the Fire Weather Index (inspired from the figure at
(https://cwfis.cfs.nrcan.gc.ca/background/summary/fwi)). The drying factor is an adjustment to the day length used for drying
in DC, renamed here to avoid confusion with the effective day length.

Afterwards, two indices are deduced from these moisture contents. The Initial Spread Index (ISI) is an indicator for the likely
rate of fire spread, while the Buildup Index (BUI) encompasses the fuel available for combustion. Together, they are used to

deduce the FWI rating the fire intensity. It is important to note that the FWI is entirely based on atmospheric variables, only providing a sense of how likely or intense a fire would be in these conditions. Information regarding vegetation is essential to assess variables such as burned area or fire emissions.

The first published algorithm for the FWI was provided by (Simard, 1970), then continuously updated for different programming languages (Wang et al., 2015). The latter source corresponds to the same equations and parameters, and it is hereafter referred to  as the original algorithm. It has been translated into several packages, although with several adjustments, expanding the initial focus on Canadian forests for an adaptability to other regions. These adjustments make use or extend further the suggestions from (Lawson and Armitage, 2008). In this current study, we implement in python the major options for three types of adjustment described below (day length, 'drying factor' and overwintering) and test their effect applied to all latitudes and to bins of latitude in Section 3.a to 3.c. To our knowledge, the effect of such adjustments have not been published, except for overwintering (McElhinny et al., 2020). The details of the considered packages are provided in Table 1. During the calculation of the DMC, the effective day length is used, and several packages proposed alternatives to day lengths adapted to Canada and the month of the year. A longer effective day length would reduce the moisture content of slightly compacted organic layers.

During the calculation of the DC, a parameter called "day-length adjustment" is used to calculate the potential evapotranspiration, thus as a drying factor of the compact organic layers. Several packages propose to adapt this value depending on the hemisphere. To avoid confusion with the adjustment brought to effective day length, we call this parameter the drying factor.

Besides adjusting for the drying factor, (Lawson and Armitage, 2008) give details on how to overwinter the DC component to account for the effects of abnormally dry winters. The effects of dry or wet winters would not carry over up to spring in the fine fuels (FFMC) or in the moderately compacted organic layers (DMC) but may for the compact organic layers (DC). In the original algorithm (Wang et al., 2015), the moisture content of deep organic layers is almost saturated in spring, even though this should not be the case in regions with dry winters. The adjustment for overwintering uses the value of DC at the end of the fire season and the precipitation up to the start of the fire season, as defined in (McElhinny et al., 2020). The onset of the fire season is defined here when the temperature is above 12ºC for the current day and the next two days, and up to when the temperature is below 5ºC for the current day and the two former days (Wotton and Flannigan, 1993). We note that two parameters are introduced by the overwintering adjustment, which are the carry-over fraction of last fall's moisture and the effectiveness of winter precipitation in recharging moisture reserves in spring. The value of the carry-over fraction depends mostly on the local snow cover, a bare soil during winter has a fraction of 0.5, while a thick cover would increase it to 1. Because we do not have this information from the ESMs, we use 0.75 as a default value everywhere. The second parameter, the effectiveness of winter precipitation, depends mostly on the soil type: well-drained soils would favour percolation and runoff, thus with a low fraction of 0.5, while poorly drained soils would be more efficient with a fraction of 0.9. Similarly, without this information from the ESMs, we use 0.75 as a default value everywhere. This adjustment is initially developed for Northern latitudes (Van Wagner, 1987), hence the terms of fall, winter and spring, but it is applied here for all latitudes.

170     The three moistures codes keep track of the past climate, changing every day the values of DMC, DC and FFMC with the current weather and the values on the former day. It implies that the scenarios are initialized with their corresponding historical, same ESM and same ensemble members. The full time series are used for calculation of the FWI, without any interruption out of the fire season, to conserve the full continuity. For the historical period, the values are initialized using the proposed method in (Lawson and Armitage, 2008).

175

| Adjustment | Method | | Package | | | | |
|---|---|---|---|---|---|---|---|
| | **Name** | **Description** | original | pyfwi | NCAR | cffdrs | current study |
| Effective day length (DMC) | *original* | Values depending on the month | X | | | | X |
| | *bins of lat.* | Bands of latitude (°N): [-90, -30]; [-30, 0]; [0, 30]; [30, 90] | | X | | | X |
| | | Bands of latitude (°N): [-90, -30]; [-30, -10]; [-10,10]; [10, 30]; [30, 90] | | | | X | |
| | *continuous lat.* | Function of latitude of grid cell and day of year | | | X | | X |
| Drying factor (DC) | *original* | Values depending on the month | X | | X | | X |
| | *two hemi.* | Depends as well on the hemisphere: Northern identical and Southern shifted by 6 months | | X | | | X |
| | *two hemi. & tropics* | Bands of latitude (°N): identical for [20, 90], shift of 6 months for [-90, -20], average for [-20, 20] | | | | X | X |
| Overwintering (DC) | *original* | No | X | | | | X |
| | *overwintering* | Yes | | | | X | X |

**Table 1.** Details of the adjustments for each one of the codes considered. The codes for each package are available online (Wang et al., 2015; pyfwi; NCAR; cffdrs). The drying factor is an adjustment to the day length used for drying in DC, renamed here to avoid confusion with the effective day length. Options in bold are those used for the data provided by this study.

## 2.3. Database features

After calculation of the FWI on the native grid of the ESM, we then use second order conservative remapping (Jones, 1999; Brunner et al., 2020) to regrid them onto a common 2.5° x 2.5° longitude-latitude grid to enable comparison across different models.

Annual indicators are made available and presented in Section 3 to illustrate these data and ease its interpretation. We use the four following annual indicators, all of them defined in (Abatzoglou et al., 2019; Jolly et al., 2015) although with a reference period of 1850-1900 instead of 1861-1910:

- Extreme value of the FWI (*fwixx*): local annual maximum value of the FWI
- Number of days with extreme fire weather (*fwixd*): local annual number of days above the local threshold defined as the 95-th percentile of the FWI over the reference period.
- Length of the fire season (*fwils*): local number of days above the local threshold defined as the mid-range of the extrema in FWI over the reference period.
- Seasonal average of the FWI (*fwisa*): local annual maximum of the 90-day running average of the FWI.

Here, *fwixd* uses a definition of what is an "extreme" fire weather based on the 95-th percentile like in (Abatzoglou et al., 2019), and not an absolute set of classes. This approach generalizes the method for analysing the FWI which attributes fire danger classes (e.g., "very low", "low", "moderate", "high", etc) to intervals of values. These classes are rather used for local or regional cases, hence defined for the region, such as China (Tian et al., 2011), Europe (San-Miguel-Ayanz et al., 2022), Greece (Varela et al., 2018), Ontario (Martell, 2000) and Malaysia & Indonesia (Dymond et al., 2005; de Groot et al., 2007). The proposed procedure to define these classes is to assume how many extreme days should be allowed on average for each season and deduce the threshold for the "extreme" class from this assumption and a sample of the FWI over a reference period. In our case, we cover the whole Earth, and the "extreme" fire danger class needs a consistent definition. By assuming a local threshold based on the 95-th percentile, it is consistent with the assumption of about 18 days per year that are considered locally as extreme fire weather. No other fire danger classes are used by our other annual indicators.

As a remark, there are two definitions for the fire season, which are kept for consistency reasons. The adjustment for overwintering DC uses a definition of the fire season based on temperature thresholds as described in (Wotton and Flannigan, 1993). However, accordingly to (Abatzoglou et al., 2019), the annual indicator *fwils* defines the fire season when the daily FWI is above a local threshold defined as the average of the minimum and the maximum of the FWI in the reference period. Given that the FWI is calculated only using atmospheric variables, regardless of the vegetation cover, we mask the results according to ESA CCI land cover of 2016 (ESA-CCI, 2017, 2019). Similarly to (Abatzoglou et al., 2019), when more than 80% of the surface of the grid cell is flagged as bare areas, water, snow & ice or sparsely vegetated, it is considered as infrequent burning.

No bias correction is applied here, as it is not within the scope of this paper, and because the method may depend on the intended application of the FWI. One may decide to correct the FWI through adjustments in the four inputs of the algorithm

via various possible ways to account for climate model biases (François et al., 2020), while others may prefer to correct the FWI itself through observations-based FWI (Field et al., 2015; Field, 2020).


## 3. Results

### 3.1. Sensitivity to the adjustments on the day length

As indicated in Table 1, several adjustments are introduced to the effective day length used for the calculation of the DMC. We show the effect of these adjustments in Figure 3 with example maps for the July 1 2014 and climatologies. Figure A.2

replicates Figure 3, albeit for the January 1, 2014. The DC and FFMC are not represented, because they are not affected by this factor. The adjustments mostly change values in the Southern Hemisphere, where the effective day lengths were not prepared in the original calibration. For instance, in December-January, when the fire season is active in Southern land, a longer effective day length means a slightly drier organic layer. As moisture content decreases, the DMC values increase and so does the FWI. We note that using the NCAR day length function (NCAR) tends to extend the day length, thus increasing

the FWI in Northern land as well.

For the first adjusted version (pyfwi), the day is shorter below 30°N, and thus the drying is lower (Fig. 3 c-f). In the Southern land, this version increases the DMC by up to 62% in December-January and decreases the DMC by -38% in June-July (Fig. 3k). In 2081-2100 of *ssp585*, these differences change respectively to 60% and -36%, then with just a slight reduction of the range (Fig. 3k). This reduction might be due to the stronger drying regime at the end of ssp585 compared to the historical

period, causing this adjustment affecting drying not to matter as much. With a lower DMC, the BUI is reduced, thus decreasing the FWI (Fig. 3l). The FWI is then increased by up to 14% in December-January and decreased by up to -14% in June-July. The range of the change in FWI changes to 15% and -10% in 2081-2100 of *ssp585*. Tropical and Northern land are less affected because the magnitude of the adjustment to the effective day length is smaller (Fig. 3 g-j).

For the second adjusted version (NCAR) in the Southern land, the FWI is increased by up to 19% in December-January and

decreased by up to -6% in June-July (Fig. 3k-l). In 2081-2100 of *ssp585*, the range of these differences is changed to 21% to -5%. However, the Northern land is also affected, with an increase of the FWI by up to 10% in December and only by 1% in April.

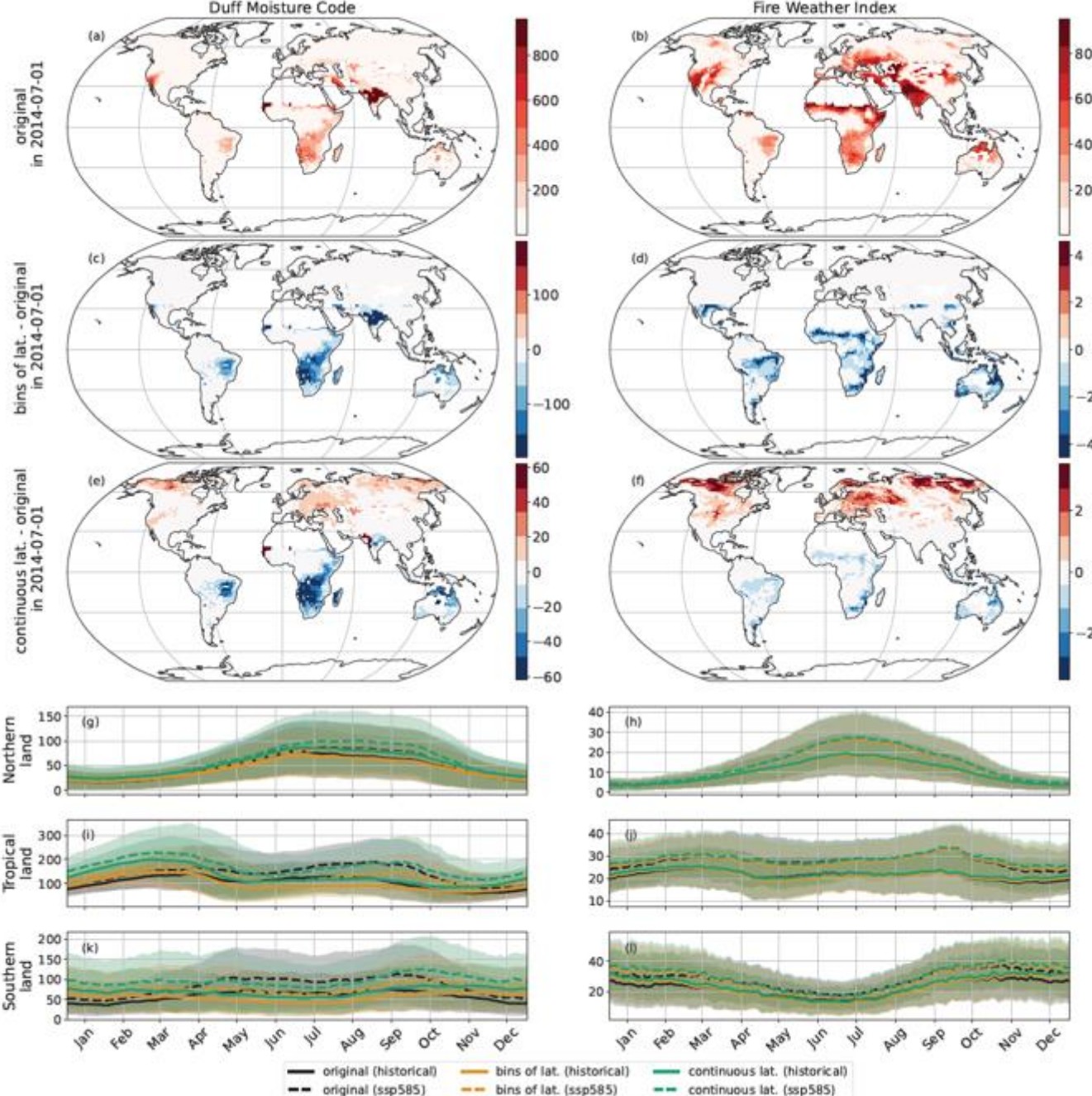

**Figure 3.** Sensitivity to the day length factor of ACCESS-CM2 in the experiments *historical* and *ssp585* over the ensemble member r1i1p1f1. The first row of maps (a & b) shows the values of the DMC and the FWI, the first and final indices affected by this factor, in the original version of the algorithm (Wang et al., 2015). The following maps (c to f) show the differences

with the adjusted version of the algorithm (Table 1). The following rows show the daily climatologies over 1995-2014 (solid line) and 2081-2100 (dashed line), represented in terms of average and ±1 standard deviation range. The Northern land (g & h) is defined as grid cells over 20°N and not marked as infrequent burning. Similarly, Southern land (i & j) is below 20°S and Tropical land (k & l) is the intermediate latitude band.

## 3.2. Sensitivity to the adjustments on the "drying factor"

The drying factor has also received several adjustments in the considered packages (Table 1 and Fig. 4). We note that the original name of this factor is "Day-length adjustment in DC" (Wang et al., 2015), but it is renamed here to avoid confusion with the adjustments brought to the "Effective day length". Figure 4 shows the effect of the adjustments brought to this drying factor through maps on July 1 2014 and climatologies, while Figure A.3 shows its equivalent on January 1, 2014. The DMC

and FFMC are not represented here, because they are not affected by this factor.

As described in Table 1, the original code proposes a single monthly value over the Earth, while the first adjustment (pyfwi) proposes one monthly value for each hemisphere, and the last one (NCAR) goes further by adding one constant value between 20°S and 20°N. In Figure 4, this is why this adjustment has mostly an impact below 20°S and to a lesser degree between 20°S and 20°N. For this reason, and similarly to Figure 3, we mainly discuss here the effect in the Southern land, which is identical

in the two adjusted versions. For the July 1 2014, DC is lower in the adjusted versions (Figure 4). According to the algorithm, the drying factor is lower in July at these latitudes, causing a lower potential evapotranspiration, consistent with a higher moisture content and a lower DC.

The major differences for the DC in the Southern land, below 20°S, are an increase by up to 43% in February-March and a decrease by up to -38% in July-September. The adjustment to the drying factor in the Southern land is at its peak in July and

at its minimum in January, one month before the observed maxima in the differences in DC. With a lower DC, the BUI is also decreased, causing the FWI to decrease. In the Southern land, the FWI increases by up to 4% in February-March and decreases by up to -4% in July-September.

This adjustment has a smaller impact on FWI than on DC and is also relatively lower compared to the effect of the adjustment on the effective day length. However, sensitivities of FWI are by increasing order to FFMC, then DMC and finally DC (Dowdy

et al., 2010), mostly because a fire starts with the least compact organic layers to move to the most compact ones. Even though this adjustment has a relatively low impact, for latitudes below 20°S, these adjustments help in adapting the climate effects on the most compact organic layers, which is of interest to reproduce seasonal cycles and long term effects of climate change (Van Wagner, 1987).

## 3.3. Sensitivity analysis to the overwintering

Overwintering the DC was suggested in (Lawson and Armitage, 2008), to account for the effects of abnormally dry winters as explained in Section 2b. This adjustment affects mostly the DC above 20°N, because of the climatological conditions under these latitudes. The stronger impact can be seen during the Northern winter. Figure 5 illustrates the effect of this adjustment for the July 1 2014 and climatologies, while Figure A.4 does so for the January 1, 2014. As expected, this adjustment affects mostly the DC above 20°N, the winters below 20°N being not dry enough, except in the South of Argentina and Chile.

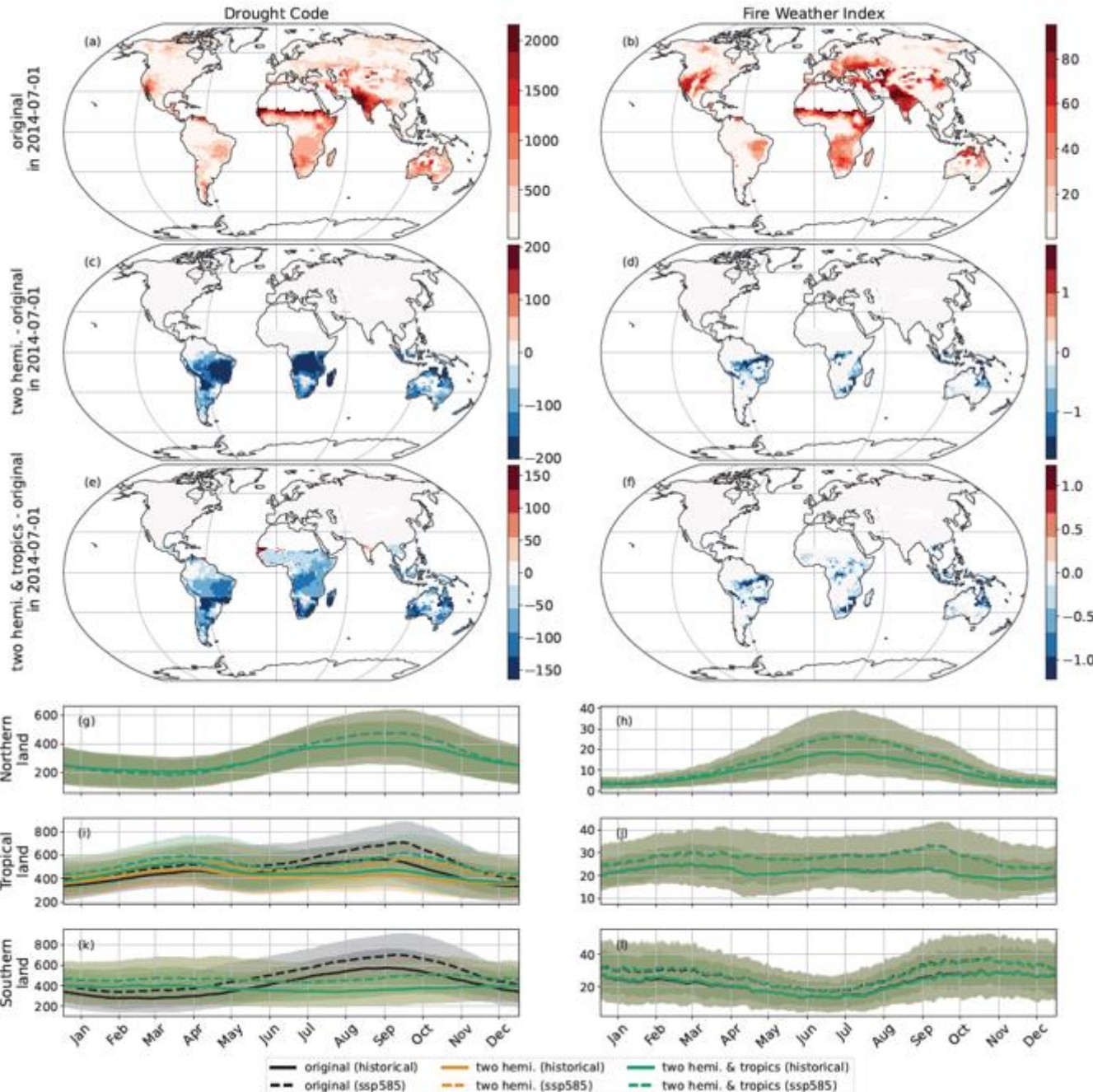

**Figure 4.** Sensitivity to the drying factor of ACCESS-CM2 in the experiments *historical* and *ssp585* over the ensemble member r1i1p1f1. The first row of maps (a & b) shows the values of the DC and the FWI, the first and final indices affected by this factor, in the original version of the algorithm (Wang et al., 2015). The following maps (c to f) show the differences with the

adjusted version of the algorithm (Table 1). The following rows show the daily climatologies over 1995-2014 (solid line) and 2081-2100 (dashed line), represented in terms of average and ±1 standard deviation range. The Northern land (g & h) is defined as grid cells over 20°N and not marked as infrequent burning. Similarly, Southern land (i & j) is below 20°S and Tropical land (k & l) is the intermediate latitude band.


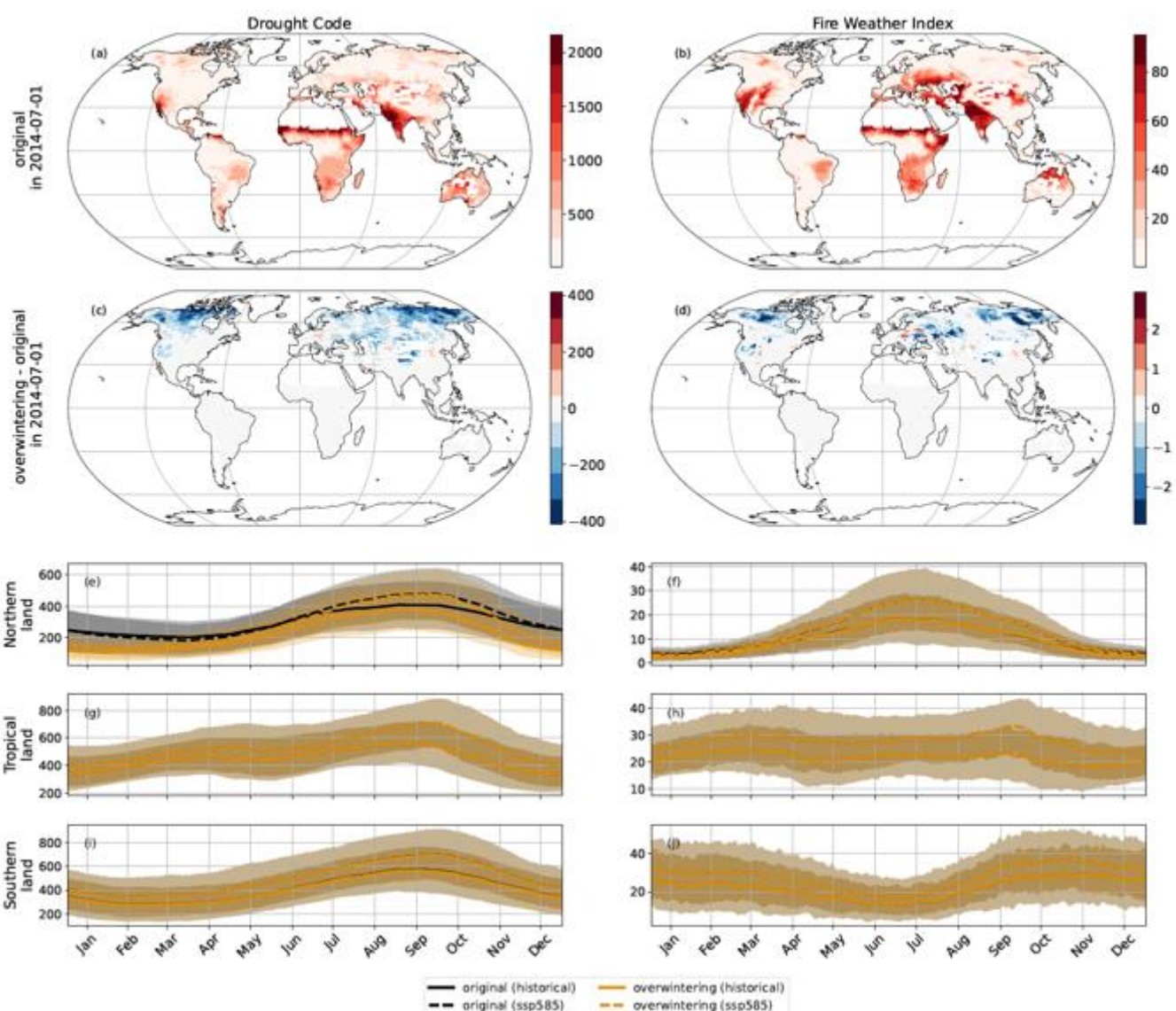

**Figure 5.** Sensitivity to the overwintering of ACCESS-CM2 in the experiments *historical* and *ssp585* over the ensemble
member r1i1p1f1. The first row of maps (a & b) shows the values of the DC and the FWI, the first and final indices affected

by this factor, in the original version of the algorithm (Wang et al., 2015). The following maps (c & d) show the differences with the adjusted version of the algorithm (Table 1). The following rows show the daily climatologies over 1995-2014 and 2081-2100, represented in terms of average and ±1 standard deviation range. The Northern land (e & f) is defined as grid cells over 20°N and not marked as infrequent burning. Similarly, Southern land (g & h) is below 20°S and Tropical land (i & j) is the intermediate latitude band.

We observe a reduction in DC by up to -55% in December-January in Northern land. The DC is decreased over the full year, the minimum of this reduction being -6% in July-September. By definition, lower DC means higher moisture content of compact organic layers. Normally, the moisture content after a dry winter should be lower than after an average winter. In the default algorithm, the calculation of the FWI stops during winter and resumes in spring. To initialize the calculation in spring, DC uses the default value of 15 as a saturated moisture content (Van Wagner 1987). This leads to an overestimation of the moisture content in spring, especially after dry winters. Similar to (Abatzoglou et al., 2019), in the version called "original", we run the full time series, without interruption in winter, meaning no initiation of the DC with a saturated level. In Figure 5, we see that adding overwintering increases the moisture content in winter and spring. It implies that running the full time serie with the default code tends to overestimate the drying during winter. In this sense, this is consistent with (McElhinny et al., 2020). The added value of overwintering is to balance the overestimation of spring moisture content if interrupting calculation of the FWI, or the underestimation of spring moisture content in uninterrupted calculation of the FWI.

With too dry moisture content in the compact organic layers, the FWI in the Northern land tends to be higher in the original version (which runs the code throughout the year) relative to the overwintered version. Overwintering reduces the FWI by up to -18% during January-February and brings an important adjustment to DC. Calculating the full time series of FWI means not re-using the saturated starting value for spring DC, causing moisture content over the year to be too low, hence the DC too high. We consider that overwintering is necessary when adjusting this effect in full time series.

### 3.4. Sensitivity to using daily mean relative humidity

The default approach would be to use as inputs daily minimum relative humidity and daily maximum temperature. As explained in Section 2.a, here we use daily mean relative humidity as an alternative, because daily minimum relative humidity is not provided for many CMIP6 runs, reducing the total number of runs from 1486 to 1321. We show in Figure 6 the influence of this choice, with climatologies and maps for the July 1 2014. Figure A.5 extends this figure using the maps for the January 1, 2014.

The DC takes as climate inputs only temperature and precipitation, but not relative humidity. This component is therefore not affected by this choice. Using daily mean relative humidity instead of the daily minimum relative humidity increases the moisture content of the other two components, as evidenced by lower DMC and FFMC. For DMC, it is reduced above 20°N by -25% (January) to -21% (October-November). In the Tropical land, between 20°N and 20°S, this reduction changes to -

15% (March-April) to -28% (June-July). Finally, below 20°S, the DMC is reduced by -34% (June-July) to -20% (November-
December). The reductions tend to be stronger for FFMC, ranging from to -45% in June below 20°N and -22.7% between
20°S and 20°N. With a more humid climate as input, and consequent lower DMC and FFMC, the FWI itself is reduced. During
the fire season, the FWI is reduced by about -33% (June) above 20°N, -35% between 20°S and 20°N (May) and -29%
(December) below 20°S. Over 2081-2100 of *ssp585*, these reductions of the FWI are changed to -31% (June) above 20°N, -
30% (June) between 20°S and 20°N and -25% (November) below 20°S.

We note that the reduction in FWI caused by using mean relative humidity instead of minimum relative humidity is roughly
identical during the fire seasons across the planet, be it over 1994-2014 or 2081-2100. It implies that analysis based on relative
changes in the FWI would not be strongly affected by this choice. As discussed in Section 2.a, the annual indicators are
provided for both cases, i.e., using daily mean relative humidity and using daily minimum relative humidity.

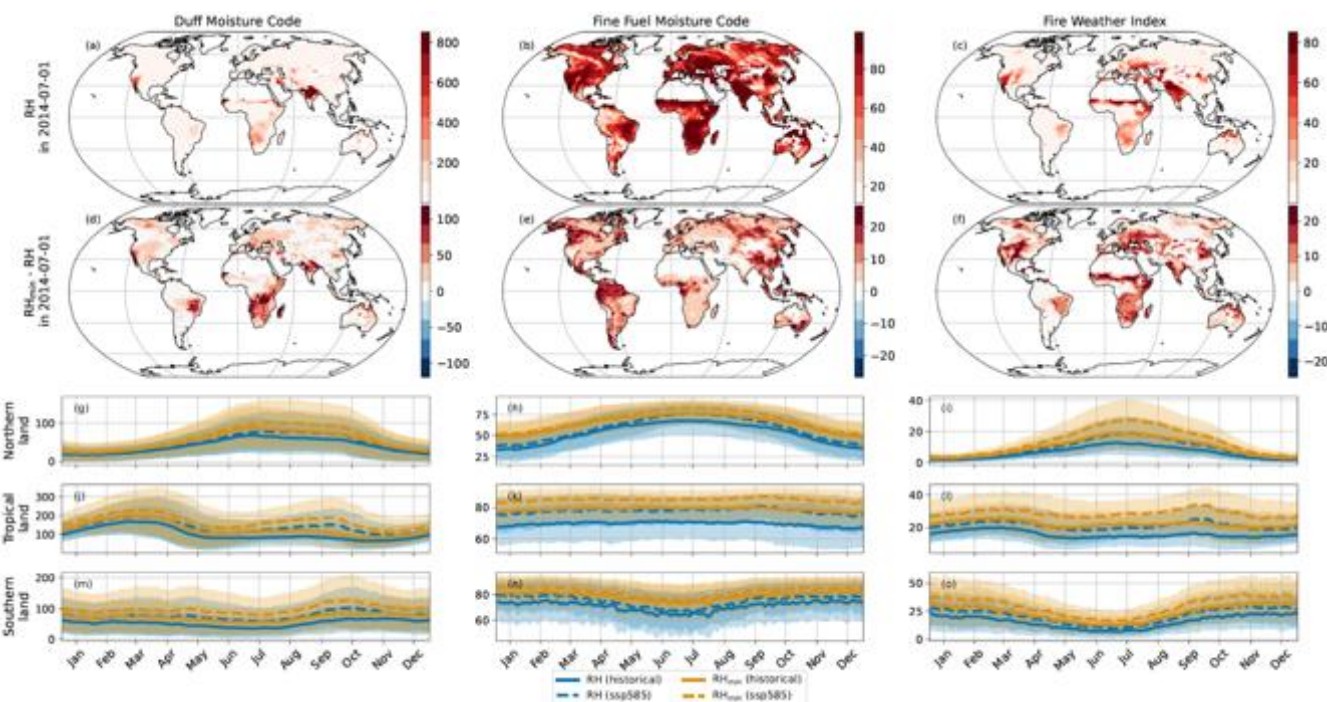


**Figure 6.** Sensitivity to the variable used for daily relative humidity of ACCESS-CM2 in the experiments *historical* and *ssp585*
over the ensemble member r1i1p1f1. The first row of maps (a to c) shows the values of the DMC, the FFMC and the FWI, the
first and final indices affected when run with daily average relative humidity. The following maps (d to f) show the differences
when run with daily minimum relative humidity. The following rows show the daily climatologies over 1995-2014 and 2081-
2100, represented in terms of average and ±1 standard deviation range. The Northern land (g to i) is defined as grid cells over
20°N and not marked as infrequent burning. Similarly, Southern land (j to l) is below 20°S and Tropical land (m to o) is the
intermediate latitude band.

### 3.5. Main results

As illustrated in Figure 1, the FWI was calculated for a total of 1321 runs, deduced from 28 ESMs, run over the historical period (1850-2014) and over 8 scenarios (2015-2100), and with a total of 108 different ensemble members. To synthesize these gridded daily products, annual indicators are provided, as detailed in Section 2.c. To summarize these annual indicators, we choose to show their evolutions at different global warming levels (GWLs) in Figures 7 and 8, while representing the robustness of the signal. Figure 7 shows the average of the runs at each GWL, and Figure 8 shows the 90[th] percentile. A signal

is defined as robust where at least 80% of the models agree on the sign of the change with reference to 1851-1900. More details on the method used to synthesize these data are provided in Text A.1 of the Appendix.

In Figure 7, we observe that all robust signals in the annual indicators of FWI show an increase. In other words, there is nowhere on Earth where ESMs agree at 80% or more on a decrease in fire weather, in any of the four annual indicators represented here. For most regions of Earth and incremental warming levels, all the four annual indicators show an increase in

their average, even if this signal is not robust in all regions. Still, there are non-robust decreasing trends in the average in Central Africa, India, and North of the Tibetan Plateau. It concerns the length of the fire season, the annual maxima, and the seasonal average of the FWI, but not the number of days with extreme fire weather that continue to show an increasing trend in these regions. The regions with robust signals are Central North America, Northern South America, Europe, Southern Africa, and Australia. We note that the higher the GWL and the larger is the spatial extent of the robustness of the signal across regions.


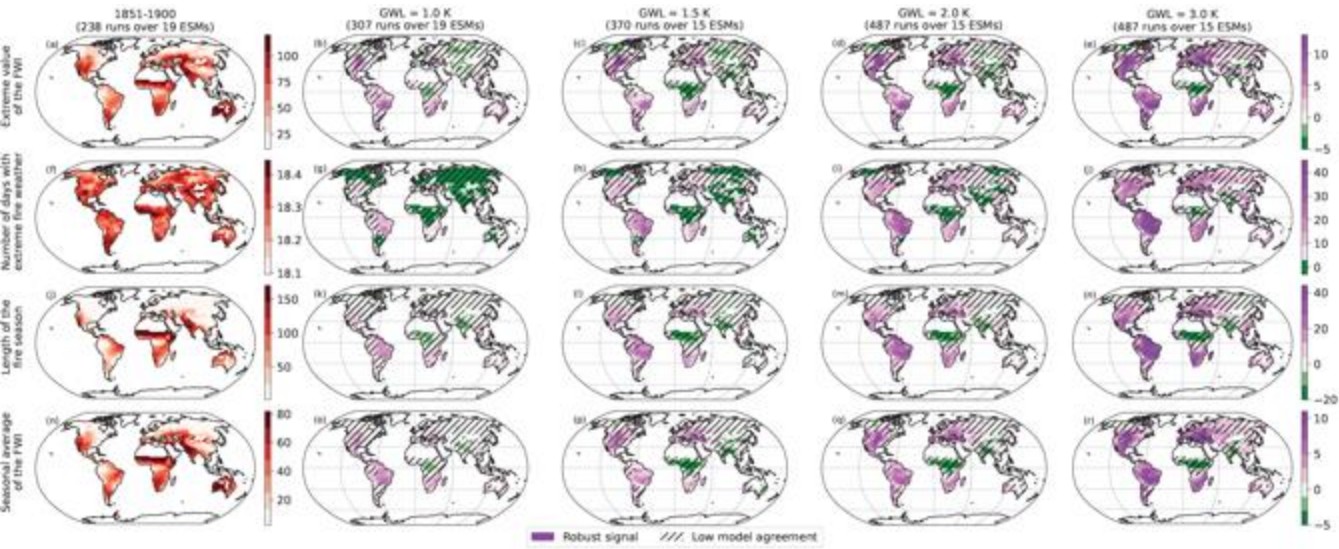

**Figure 7.** Averages of annual indicators at preindustrial and changes in averages at different Global Warming Levels (GWLs). The average maps in 1851-1900 are shown in (a), (f), (j) and (n) for the four annual indicators. Each column corresponds to

the changes in averages at different GWL, each row corresponding to a different annual indicator. The agreement of models on the sign of the change is used to define the robustness of the signal. More details are provided in Text A.1.

From Figure 7, we infer that the annual maximum of the FWI has increased by up to +37% with a GWL of 1.0°C. With a GWL of 3.0°C, this indicator is increased by up to +120%, its average being +9%. Western Australia exhibits the highest

robust trend for this indicator. The number of days with extreme fire weather has increased by up to +128% with a GWL of 1.0°C. With a GWL of 3.0°C, it increases by up to a factor of 5.0, while its average is +86%. South America would be the most affected with 48 days of extreme fire weather per year, while the world would have on average 26 days. The length of the fire season increases as well, even at a GWL of 1.0°C, up to a factor of 4.0. With a GWL of 3.0°C, it may increase by a factor of 15.0 and on average +66%. Such a high relative increase happens in Northern Siberia, where the statistical distribution

of the daily FWI and the definition of this indicator lead to an extremely short fire season. Nevertheless, the average increase by +66% shows that the fire season is overall increasing, lasting about 45 days on average. At a GWL of 1.0°C, the seasonal average of the FWI is also increasing, by up to 56%. However, at a GWL of 3.0°C, it increases by up to +250% and on average by +17%. Indonesia would be the most affected region according to this criterion.

Overall, these annual indicators emphasize that under climate change, atmospheric conditions tend to increase the number and

intensity of fires. An approximate estimate is that we would expect at least an average +66% in both the frequency and the duration of fires at a GWL of 3.0°C relative to 1851-1900, although this is highly dependent on the metric and the region.

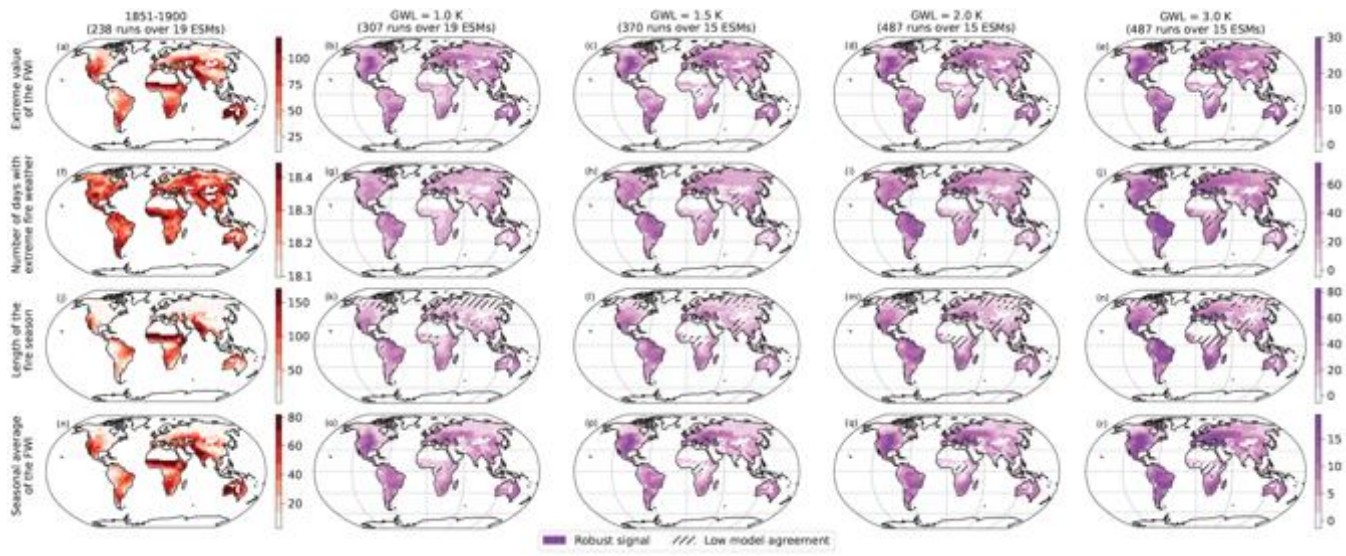

 **Figure 8.** 90% percentiles of annual indicators at preindustrial and changes in averages at different Global Warming Levels (GWLs). The average maps in 1851-1900 are shown in (a), (f), (j) and (n) for the four annual indicators. Each column corresponds to the changes in 90% percentiles at different GWL, each row corresponding to a different annual indicator. The agreement of models on the sign of the change is used to define the robustness of the signal. More details are provided in Text A.1.

In analogous fashion to Figure 7, we show the FWI indicators of the 90th percentile in Figure 8. The 20-year window centred at time of exceedance of the GWL is used here to deduce the 90th percentile instead of the average. It assumes that over the 20 years at each GWL, the distributions of the annual statistics of the FWI are stationary enough such that the contribution of the local warming trend is lower than the local natural variability. Under this assumption, the percentiles of all runs exceeding the GWL provide the local 1-in-10-year event and the local model agreement. More details are provided in Text A.1 of the Appendix. In Figure 7, all robust signals are increasing trends, while decreasing trends are always non-robust signals. In Figure 8, all signals, robust or not, are increasing trends. Regarding the robustness of the signal, we notice that the area extent of the robust signals is higher for the 90% percentile than for the average. In other words, even if ESMs may disagree in some regions on the sign of the average change in annual indicators of FWI, they agree much more that 1-in-10-years fires will be higher. As a summary for these trends, at a GWL of 3.0°C, we calculate average relative increase of the 1-in-10-years events of +31% for the annual maximum of the FWI, +192% for the number of days with extreme fire weather, +177% for the length of the fire season and +46% of the seasonal average. Overall, the 1-in-10-years events would thus almost triple in duration and increase by at least 31% in strength. These findings highlight how fire management, even more than nowadays, would increasingly become an absolutely crucial element of forest protection across the world. However, there would be some limits to the adaptation possible to these projected changes, highlighting that the best course of action would require limiting global warming as low as possible.

## 4. Data availability

The fire weather index data has been generated using data archived on the ETH Zurich CMIP6 repository. CMIP6 model outputs can also be accessed through different Earth System Grid Federation (ESGF) data nodes. The fire weather indices produced in this paper is made available in netCDF format and it is openly and anonymously accessible through https://doi.org/10.3929/ethz-b-000583391 (Quilcaille and Batibeniz, 2022). All files are stored as .zip in the archive. They are named as "[Long name of the fire weather indicator] for all available CMIP6 runs (computed using [Long name of the humidity used for calculation])". Each .zip file contains the results for this indicator for different ESMs, scenarios and ensemble members. For storage reasons, some of them are stored in several parts. Within each of .zip files, the results for each fire weather indices, ESM, scenario and ensemble member are saved individually in separate files in a netCDF4 format under the

name "[indicator]_ann_[ESM]_[scenario]_[member]_g025.nc". Here, "ann" designs the annual resolution while "g025" designs the name of the grid.

## 5. Code availability

The fire weather index results have been generated using the open-source model developed by YQ and improved by YQ and FB. The code to reproduce the results of this manuscript is available on a repository on GitHub (https://github.com/yquilcaille/FWI_CMIP6).

## 6. Usage notes

The provided data is produced by the Institute of Atmospheric and Climate Science Institute of ETH Zurich. It is an open source and entirely free dataset. To illustrate possible paths for data users, we indicate in the following list some of the many opportunities where this dataset could be used. Some may rather be considered as research questions while some other points may be of interest for societal issues regarding fires.

As detailed in Section 2, we highlight that CMIP6 data may come with biases, while observations provide more realistic inputs

and information for fire related studies. Though, observations have lower temporal and spatial availability and cover only the historical period. Thus model-based data facilitates large scale analysis.

- Comparison of FWI results with observations to evaluate the biases in the models. Compared to observations, some models show biases in their outputs. How does that affect the calculation of a compound product like the FWI? The FWI can be calculated using data based either on models or on observations (e.g. (Vitolo et al., 2019)). One may use

the dataset provided here to evaluate the discrepancies and eventually how it affects future projections in fire weather. A first work in this direction has been produced with 16 ESMs and 1 ensemble member over the historical period (Gallo et al., 2022).

- Discrepancies in FWI across ESMs projections. The ESMs show different regional evolutions in some variables, though the effect of these discrepancies on the FWI remains unclear. One may investigate how much do projections

in fire weather depend on the ESM by using the provided dataset and investigate reasons for the (dis)agreements.

- Dependencies of the FWI to ensemble members. The former path could be extended to the ensemble members. An uncertainty in the projections of the FWI arises from the initial conditions as well. The provided dataset may be used to assess this uncertainty and eventually the natural variability in FWI.

- Dependencies of the FWI to scenarios. Another dimension of projections in the FWI is the choice of the scenario.

Under low warming scenarios, the Earth system gets more time to stabilize, allowing for different regimes, e.g. in the water cycle. It may help to investigate the response of the fire regimes across different scenarios. For example, the

differences between low warming or high warming scenarios or even overshoot scenarios can be investigated using the provided dataset.

- It can be used to understand the effects of humidity regimes on fire regimes: minimum relative humidity and average relative humidity have different dynamics, and it is still unclear how they may affect the dynamics of fire weather in current and future climate. The provided dataset may help in assessing these regimes and their differences.

- Comparison of climatology of FWI in preindustrial, current, and future climate. Figure 8 of this manuscript gives a brief overview of this path. What should we expect from fire weather at different levels of climate change? Such a question would be of interest to inform society for the implications of climate change, and the provided dataset may help to answer it.

- Relationship of fire weather to modelled burned area. There is literature showing the correlation between FWI and burned area(Jones et al., 2022), in spite of other relevant factors such as fire ignition. One may use the provided dataset to check in the CMIP6 ensemble whether these relationships could be improved, and how they could be used, e.g. in impact models.

- Attribution studies of FWI to anthropogenic climate change under historical and future projections. Heatwaves, droughts and other extreme events have been attributed to climate change, but only limited studies have been able to attribute fires or mega-fires to climate change. The lack of relevant data explains this reduced number of attribution studies. Thanks to this provided dataset, attribution studies may use this data to assess changes in probabilities due to climate change. Though, the provided dataset does not provide runs under the scenario "hist-nat", the historical run with only natural forcings but not anthropogenic forcings. It remains possible to use this dataset by considering pre-industrial period and current period with their corresponding natural variability.

- FWI under CMIP5 and CMIP6. The FWI has been calculated for CMIP5 runs in (Abatzoglou et al., 2019), while the provided dataset calculates the FWI for the latest CMIP6 exercise. A comparison of both datasets would allow us to identify changes in fire weather between the ESMs. Coupled to their respective burned areas, one may disentangle the causes for differences in fires under ESMs between fire modules and fire weather of the models.

## 7. Appendix A

| Fire weather index | Temp. | Rainfall | Relative humidity | Wind speed | Other | Reference |
|---|---|---|---|---|---|---|
| Munger | | | | | Days without rainfall | (Munger, 1916) |
| Nesterov | X | | | | Days without rainfall | (Nesterov, 1949) |
| Angström | X | | X | | | (Chandler et al., 1983) |
| Zdhanko | X | X | | | Dewpoint temp. | (Zhdanko, 1965) |

|  | | | | | |
|---|---|---|---|---|---|
| **GFDI** | X | | X | X | Fuel, grass condition | (McArthur, 1967) |
| **FFDI** | X | | X | X | Fuel availability | (McArthur, 1967) |
| **BI** | X | | | X | | (Baumgartner, 1967) |
| **KBDI** | X | X | | | | (Keetch and Byram, 1968) |
| **M68** | X | X | | | Vegetation condition | (Käse, 1969) |
| **Orieux** | X | X | | X | | (Orieux, 1974) |
| **FFWI** | X | X | X | | Drought index | (Fosberg, 1978) |
| **NFDRS** | X | X | X | X | Lightning, clouds | (Deeming, 1972) |
| **EMC** | X | | X | | | (Bradshaw, 1984) |
| **LFDI** | X | | X | | | (Meikle and Heine, 1987) |
| **FWI** | X | X | X | X | | (Van Wagner, 1987) |
| **I87** | X | | X | X | | (Carrega, 1991) |
| **Haines index** | X | | | | Dewpoint temp. | (Haines et al., 1983) |
| **Numerical risk** | X | | X | X | Cloud cover | (Sol, 1990) |
| **Portuguese index** | X | X | | X | Dewpoint temp. | (Goncalves and Lourenco, 1990) |
| **F index** | X | | X | X | | (Sharples et al., 2009a, b) |
| **FMI** | X | | X | | | (Sharples et al., 2009a, b) |
| **Fire danger** | X | X | X | | | (Setzer and Sismanoglu, 2012) |

**Table A1.** list of fire weather indices in the literature, extracted from (WSL; de Groot et al., 2015).


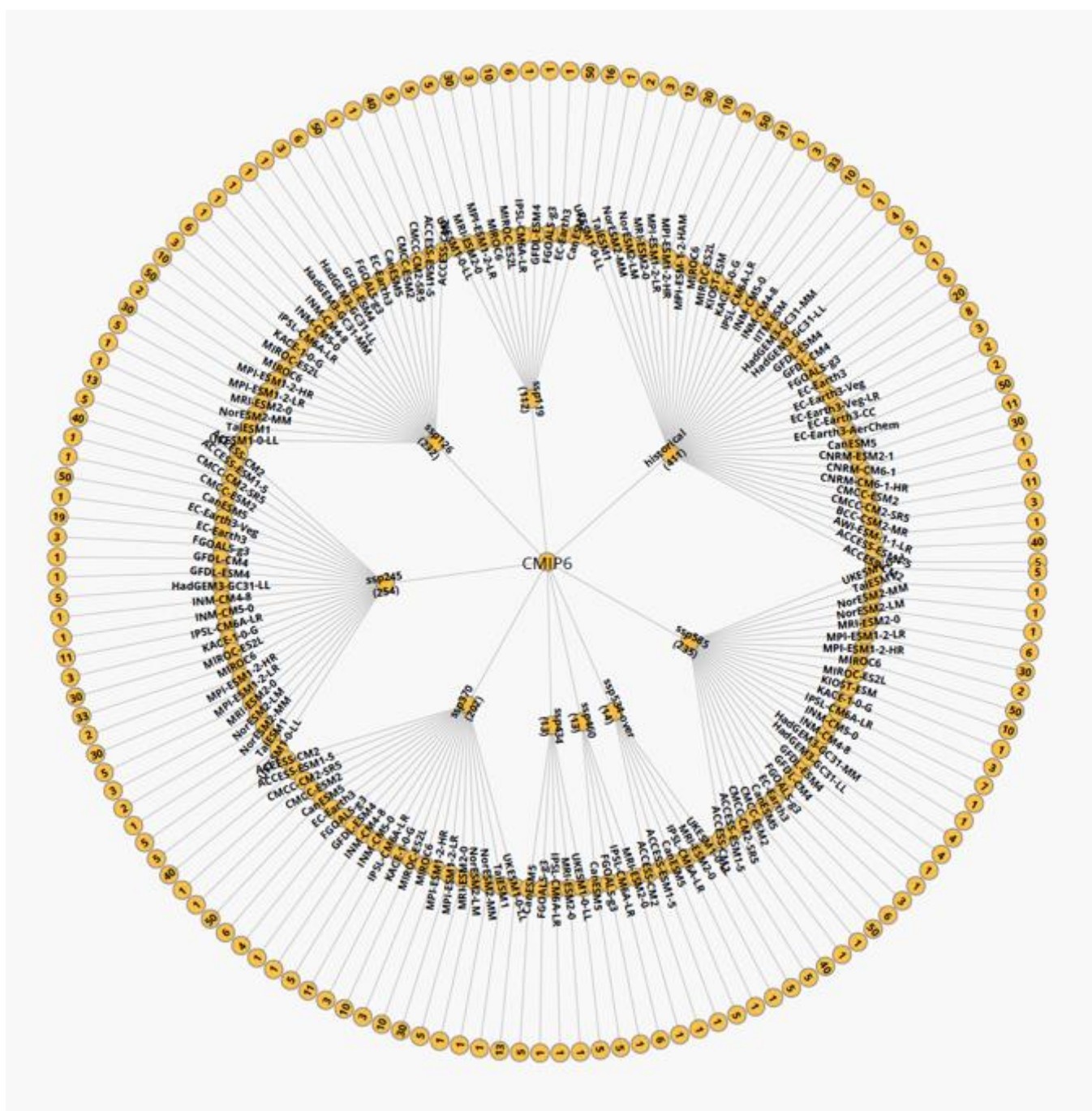

**Figure A1.** Same as Figure 1, but with runs selected with daily average relative humidity (*hurs*) instead of daily minimum relative humidity (*hursmin*). Altogether, 1486 runs are presented here.


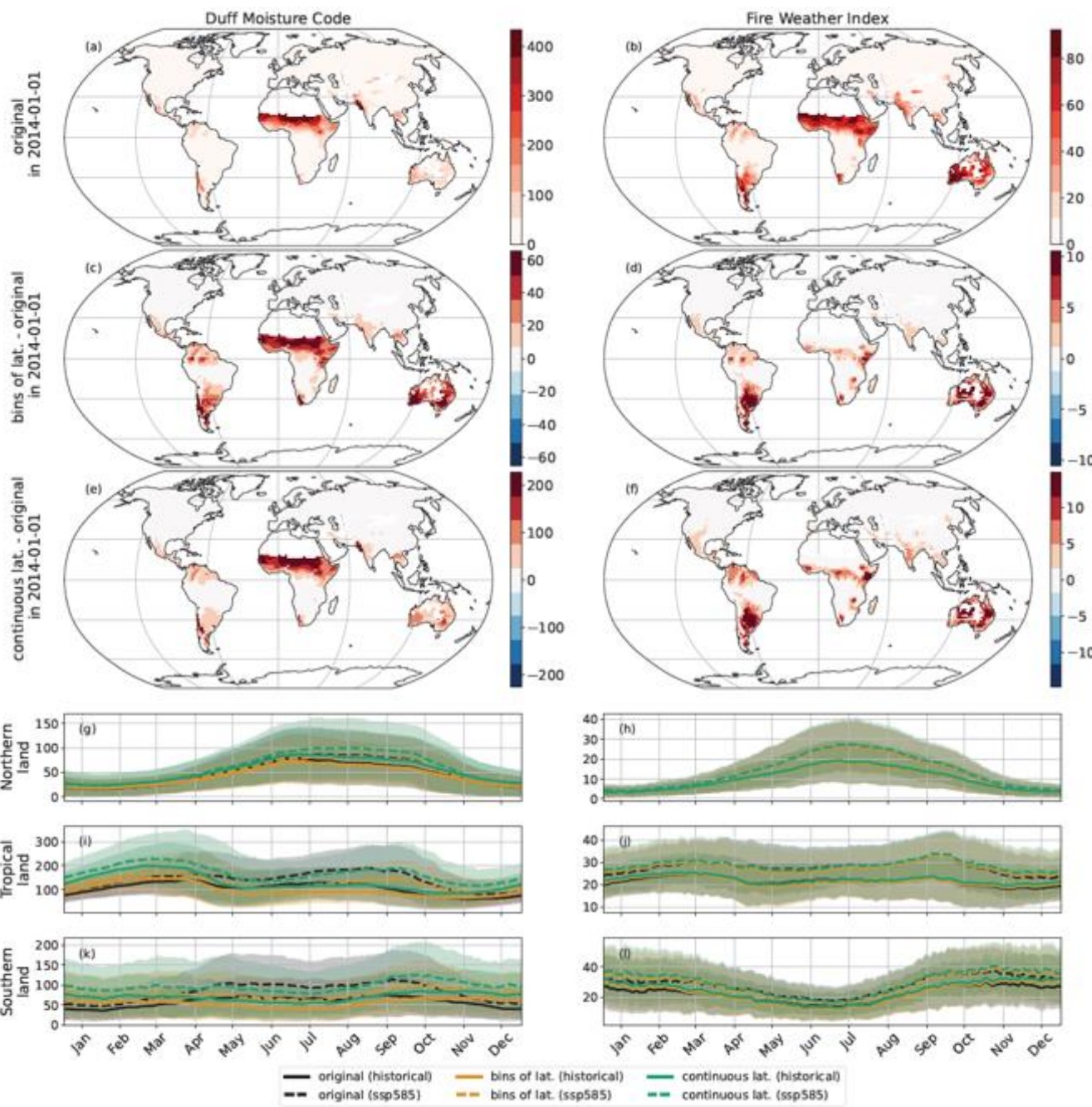

**Figure A2.** Similar to Figure 3 but on the January 1, 2014.


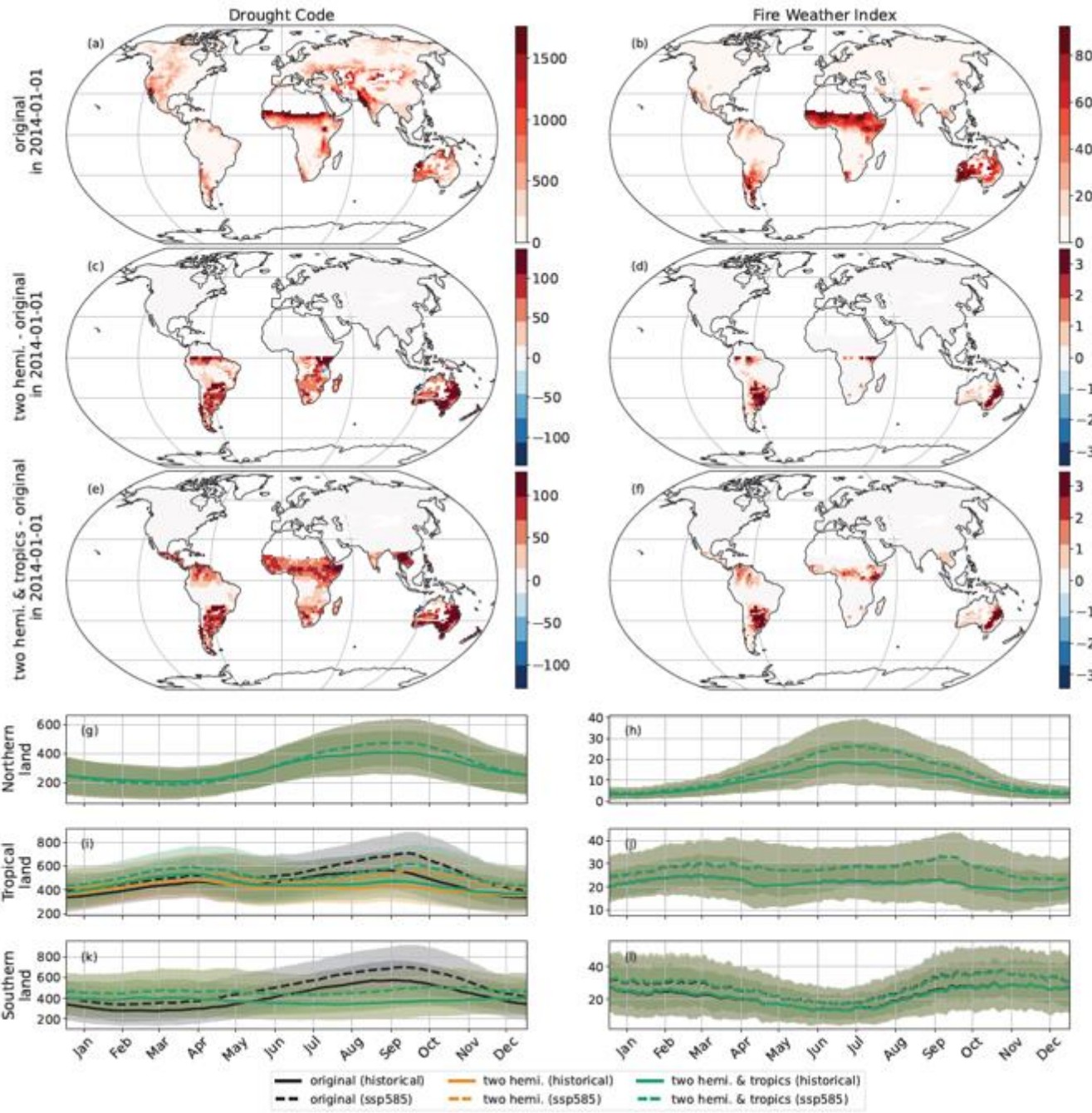

**Figure A3.** Similar to Figure 4 but on the January 1, 2014.


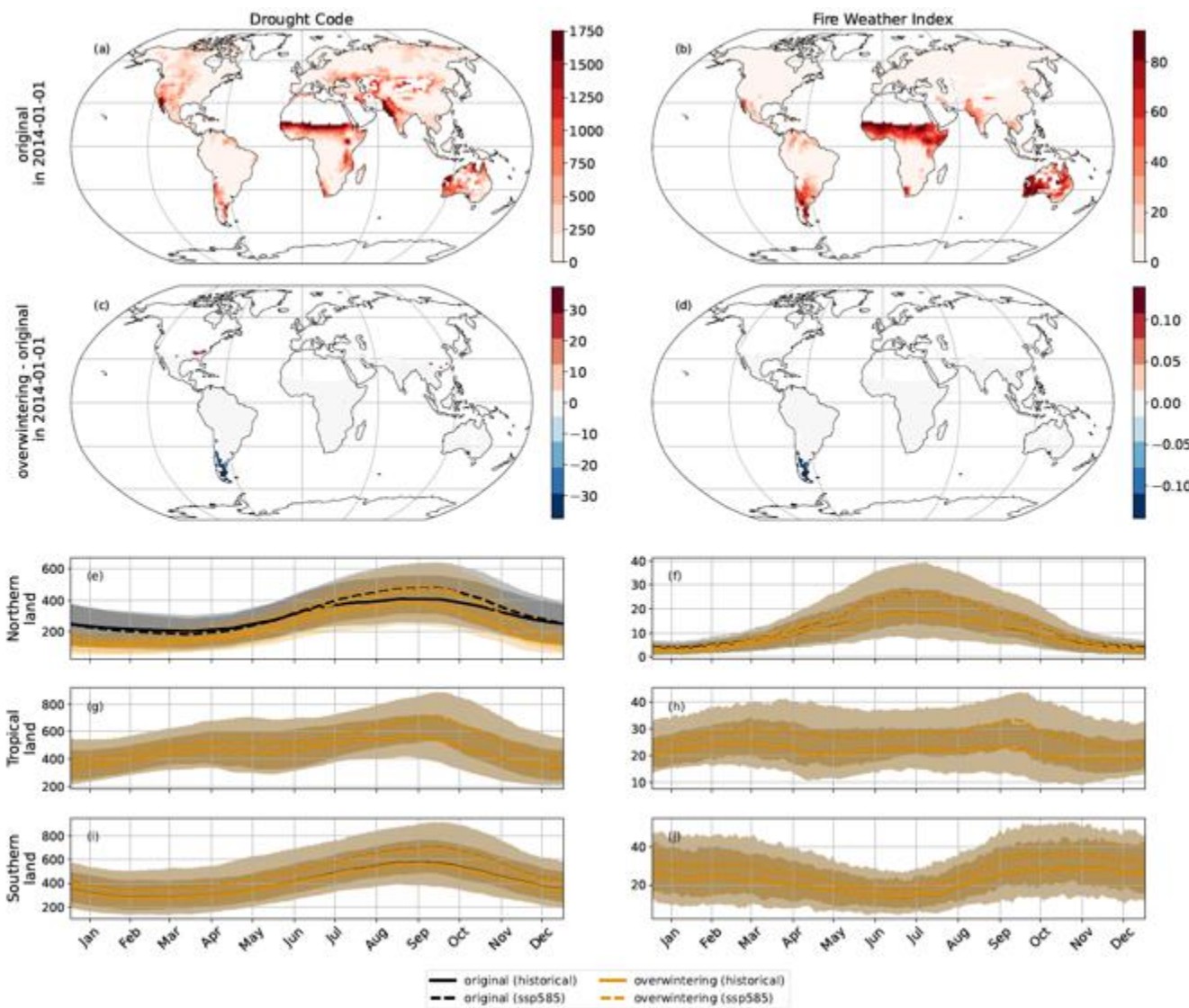

**Figure A4.** Similar to Figure 5 but on the January 1, 2014.

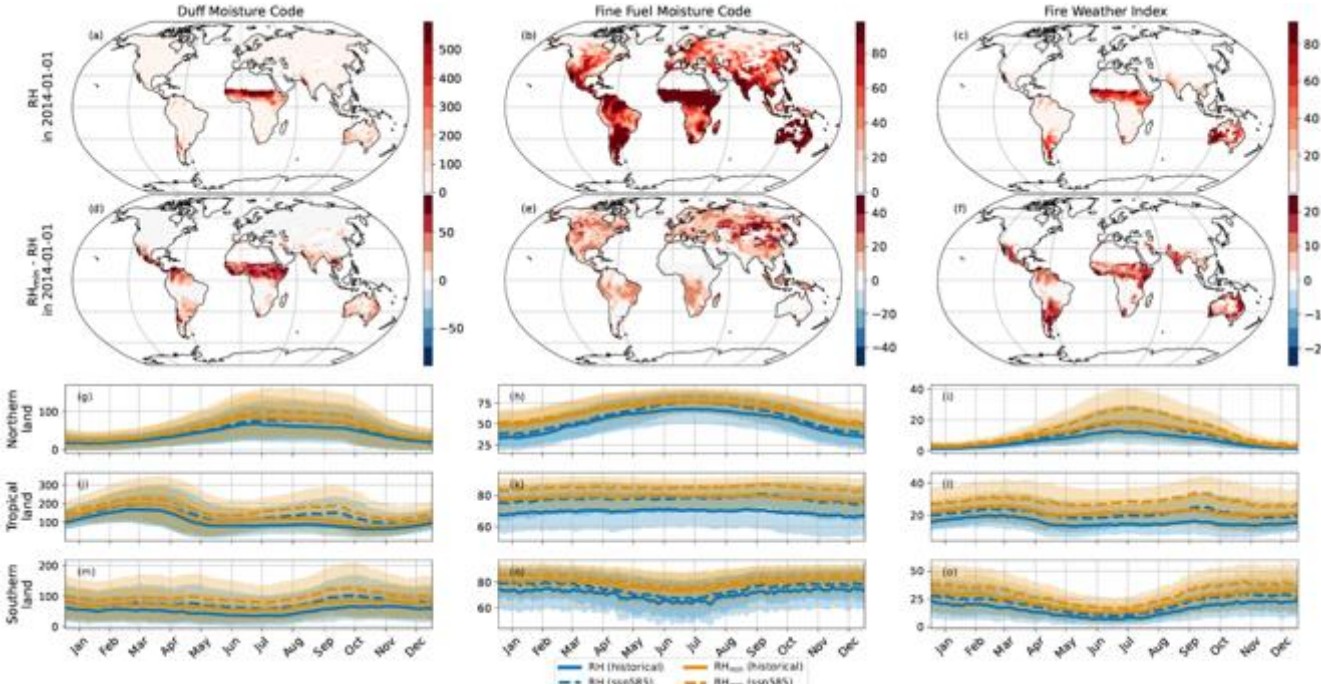

**Figure A5.** Similar to Figure 6 but on the January 1, 2014.

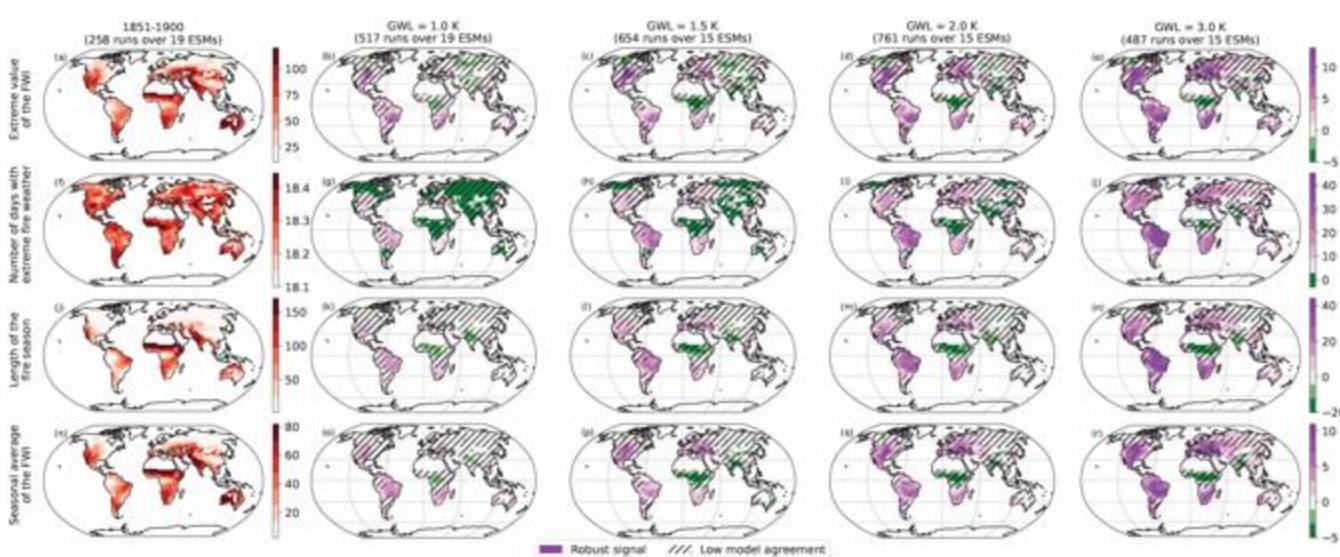

**Figure A6.** Similar to Figure 7, but with no common subset for the selection of maps at different GWLs.


**Text A1.** Method for Global Warming Levels & Uncertainties

Figures 7, 8 and A6 synthetize information on annual indicators of FWI by representing the maps at a specific Global Warming Level (GWL) and then showing their robustness. The first step is the identification of the maps of annual indicators at each GWL:

1. With 1851-1900 as reference period, we deduce the change in global mean surface temperature (*tas*) from each run used in this manuscript

2. For a given GWL, we identify for each run if the GWL is exceeded, and its first year.

3. For each annual indicator, if a run has exceeded the GWL, we gather the maps of the indicator from the exceedance year - 10 to the exceedance year + 9.

    a) If the run is a scenario with an exceedance year inferior or equal to 2023, the *historical period* is used to extend the missing year backwards.

    b) If the run is a *historical* and with an exceedance year superior or equal to 2005, the *ssp245* is chosen to extend the missing year forwards. For this reason, only runs for which a corresponding *ssp245* has been run are selected.

4. For each run, the maps over this 20 years period are averaged to obtain later Figures 7 and A6. For Figure 8, the 90% percentile is taken over this period.

5. We compile the ensemble of averaged maps reaching this GWL and proceed to map the robustness and uncertainties at each GWL.

We note that the higher the GWL, and the lower the number of runs reaching this GWL. It implies that the subset of runs used are different for each GWL. To avoid introducing a bias in the comparison, we choose to restrain the subset of runs to the runs that satisfy the higher represented GWL. It ensures the same subset of runs, although at the appropriate exceedance years. Because using this subset differs from the usual method, we have also added Figure A.6, where the full set of runs respecting the GWL are used.

The former step provides us with a subset of maps at each GWL, coming from different runs, defined by the ESM, the scenario and the ensemble member. This subset is averaged using the following method, to avoid that models with higher number of scenarios or ensemble members are over-represented. Model democracy is used here, without discarding or weighting any model.

1. At each GWL, the subset of maps is first averaged over the ensemble members, not to give more weight to ESMs with large ensembles.

2. Afterwards, the subset of maps is averaged over the experiments, not to give more weight to ESMs that run more scenarios than others.

3. Finally, we average on the ESMs.

We calculate the robustness of these maps following the approach B of IPCC (Gutiérrez et al., 2021), although the problem of over-representation of models had to be accounted for. Model democracy is used as well here.

1. At each GWL, for each ESM, the maps for available scenarios and ensemble members are pooled.

2. If more than 80% of these runs have the same sign for the evolution from the reference period, this ESM is marked as having a robust signal. As a note, the ensemble members and scenarios dimensions are not differentiated here, so that enough runs are used to evaluate the robustness of the sign.

3. If more than 80% of the ESMs at this GWL were marked as having a robust signal, then the signal is marked as overall robust.


## 8. Author contribution

YQ and FB performed FWI analyses with different methods in preparation for the manuscript and coordinated the present analyses. YQ merged the codes, computed the current database, and produced the figures and tables. YQ and FB jointly wrote the manuscript. All the authors co-designed the study, discussed the results, and contributed to the manuscript.

## 9. Competing interests

The authors declare that they do not have any competing interests.

## 10. Acknowledgements

We acknowledge the European Research Council (ERC) for ERC Proof Of Concept Grant MESMER-X (Grant Agreement ID
964013), the project PROVIDE (Grant Agreement ID 101056875) and the Swiss National Science Foundation (SNSF) through the Compound Events in a Changing Climate (CECC) (Grant agreement ID IZCOZ0_189941) project contributing to the European COST Action CA17109, "Understanding and modeling compound climate and weather events" (DAMOCLES) that both funded this work, and the project number 186282 'Risk assessment of critical ecological thresholds in Amazonia and Cerrado'. We acknowledge the World Climate Research Program's Working Group on Coupled Modeling, which is
responsible for the Coupled Model Intercomparison Project (CMIP), and we thank the climate modeling groups (listed in Figure 1) for producing and making their model output available. Furthermore, we are indebted to Urs Beyerle, Lukas Brunner, and Ruth Lorenz for downloading and curating the CMIP6 data. We are also thankful to John Abatzoglou for pointing out to the package cffdrs and the exchange on the FWI. Finally, we are thankful to Mathias Hauser for advice to curate the final products and regarding the Figures 7, 8 and A6.

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
