# Peer review of "Fire weather index data under historical and SSP projections in CMIP6 from 1850 to 2100"

_Earth System Science Data, 2022_

## Referee Comment (RC2)

The manuscript titled "Fire weather index data under historical and SSP projections in CMIP6 from 1850 to 2100" presents a global dataset of FWI changes over the long term, using all CMIP6 simulations. The data could be used to evaluate the impacts of climate change on fire danger. Although I appreciate the significant effort the authors have made in processing the CMIP6 data, calculating the FWI, and estimating the model agreement in different regions, I have some major reservations about the study in its current form, which are detailed below.

**Major Comments:**

1.  I would argue that it is a convention to use the $RH_{min}$ (rather than $RH_{mean}$) to calculate the fire weather index, as per previous studies (Van Wagner, 1987; Vitolo et al., 2019; Abatzoglou et al., 2019). Replacing RHmin with RHmean resulted in large decreases in DMC, FFMC, and FWI by 30-35% (Fig. 6). Especially, using $RH_{mean}$ influence the monthly variations of DMC and FWI, which is expected as there are much larger differences in $RH_{min}$ than $RH_{mean}$ between the wet and dry seasons. The author chose to use $RH_{mean}$ as "because daily minimum relative humidity is not provided for many CMIP6 runs, reducing the total number of runs from 1486 to 1321 (Line 303-304)". I suggest that 1321 CMIP6 runs with $RH_{min}$ would be adequate for conducting FWI predictions.

2.  The manuscript needs to highlight the novel aspects of the FWI dataset, especially compared to the one produced by Abatzoglou et al. (2019), which generally describes FWI in the same period (1860-2099). What new information could be obtained using the CMIP6 ensembles?

3.  The manuscript requires further explanation of the methods, validation, interpretation of the results, and discussions of the data limitations. Specifically, the authors need to clarify the following:

    1)  Why do we need to use the "day length" and "drying factor" adjustments? How do these adjustments play a role in different seasons? The authors need to provide equations to describe the adjustment explicitly.

    2)  How is this FWI product compared with other FWI products (Vitolo et al., 2019; Abatzoglou et al., 2019)? A comprehensive evaluation is needed in the historical period to conduct future analyses.

    3)  As we know, increases in FWI could not be translated to the burned area change directly. Then how could we interpret the increase in FWI? Please extend more discussions in section 3.5 than just providing the numbers. Also, why do we see less agreement in FWI changes in boreal Asia, boreal America, African tropical forests, and India?

    4)  Clarify the data limitations for readers who will use this data for analyses.

**Comments on figures:**

Figure 3:

    1)  I suggest that the authors show the changes in the fire season of the Southern Land which has a greater difference, for example, January 1st instead of July 1st. They could exchange Figure A2 and Figure 3.

2) Does the shaded area in Figs. (g-l) show ±1 standard deviation for historical only or both historical and SSP585. The shaded areas get so overlapped with each other and confuse me what can be learned from the figure. And there is no interpretation of this standard deviation range.

Figure 4: Again. I suggest exchanging Figure 4 and Figure A3 because the regions (Southern Land) showing large differences are in the wet season on July 1st. Therefore, we need to focus on fire season changes.

Figures 7 & 8:

1) Can you explain why you use 1851-1900 as a reference year? Is it better to use more recent years (e.g., 2000-2020) when the observations of most fire regimes are available?

2) The authors need to clarify how they calculate the number of days with extreme fire weather, the length of fire season, and the seasonal average of the FWI *at different GWLs*. For example, are you using the number of days above 95-th percentile of the FWI *over 1851-1900* to calculate *fwixd* at different GWLs?

**Minor Comments:**

1. Line 130: what are the benefits of using this day length adjustment?

2. Line 133: it would be helpful to explain how the day length parameter varies across different seasons and whether it only affects fire season.

3. Line 137: I am curious about the reason for considering potential ET. Can you provide an additional explanation?

4. Line 205: Please refrain from using "correct" here –it is unclear whether adding the adjustments would improve the FWI prediction or not

5. Line 205-210: In Fig. 3, this is a clear seasonable pattern in DMC in the Northern land and Tropical land, but not in the Southern land. Can you explain why?

6. Line 214: Why is there a decrease in the range of FWI/DMC?

7. Line 248-249: Could you rephrase the sentence where you mention "one month before the observed extreme in the differences in DC"?

8. Line 253: "However, the FWI presents higher sensitivities to changes in FFMC than to DMC, and even more to DC": This sentence is confusing: not clear if the sensitivity of FWI to DC is higher or lower than FFMC. I think Dowdy et al. show FFMC > DMC > DC. Please rephrase here.

9. Line 255: Please avoid using the word "correcting"

10. Line 343-344: "It concerns the length of the fire season, the annual maxima, and the seasonal average of the FWI, but not the number of days with extreme fire weather that continue to show an increasing trend in these regions". I still see an increase in the number of days with extreme fire weather (second row of Figure 7). Is my understanding incorrect?

Reference:

Abatzoglou, J. T., Williams, A. P., and Barbero, R.: Global Emergence of Anthropogenic Climate Change in Fire Weather Indices, Geophys Res Lett, 46, 326-336, https://doi.org/10.1029/2018GL080959, 2019.

Van Wagner, C.: Development and structure of the Canadian forest fire weather index system, 1987.

Vitolo, C., Di Giuseppe, F., Krzeminski, B., and San-Miguel-Ayanz, J.: A 1980–2018 global fire danger re-analysis dataset for the Canadian Fire Weather Indices, Scientific Data, 6, 190032, 10.1038/sdata.2019.32, 2019.

---

## Author Comment (AC1)

**Response to Anonymous Referee 1 for the manuscript**

Fire weather index data under historical and SSP projections in CMIP6 from 1850 to 2100

Yann Quilcaille[1*], Fulden Batibeniz[1*], Andreia F. S. Ribeiro[1], Ryan S. Padrón[1], Sonia I. Seneviratne[1]

[1]Institute for Atmospheric and Climate Science, Department of Environmental Systems Science, ETH Zurich, Zurich, Switzerland

*These authors contributed equally to this work

*Correspondence to*: Yann Quilcaille and Fulden Batibeniz (yann.quilcaille@env.ethz.ch and fulden.batibeniz@env.ethz.ch)

We would like to thank the reviewers for their valuable comments. We have addressed all comments of the Anonymous Referee #1, Anonymous Referee #2 and Anonymous Referee #3 through appropriate changes and hope that the revised manuscript satisfies the Referees' concerns.

The Response to the Referees file provides complete documentation of the changes made in response to each comment. While this comprehensive explanation requires some repetition of material throughout the answer, our intention is that it helps to evaluate how each comment has been addressed.

Referees' comments are shown in black. The authors' response is shown in green text. The text quoted from the manuscript is shown between quotation marks in italics. Numbers of lines correspond to the version including tracked changes.

Summary of modifications:
- Modification of abstract and introductions
- Highlighted the novelty of the dataset in abstract and introduction
- Extensive changes to the Usages Notes
- To answer concerns on RHmean vs RHmin, the paper has been rewritten to feature the data produced using RHmin as main dataset and the data produced using RHmean as secondary dataset.
    - Description of data: updated section 2.1 and Figures 1 & A1 exchanged
    - Sensitivity analysis: updated section 3.1, 3.2 & 3.3; updated Figures 3-5 and A2-A4
    - Results: updated section 3.5; updated Figures 7, 8 and A6.
    - Data: nothing changed, both datasets were already provided.
- Minor revisions in the text

Fire weather index (FWI) is an important indicator for depicting potential fire risks. When ignitions and fuel couple with higher FWI, serious wildfires could be triggered. In this study, Quilcaille et al., generated the FWI data from history and under various SSP scenarios based on the Earth System Models (ESMs) of CMIP6. They used all available ESMs with all

ensemble runs under all scenarios, therefore the generated dataset is potentially useful for understanding future changes of fire weather risks and their uncertainty. The paper is generally well written, and I only have some minor concerns listed below. I would recommend this work for publication after the concerns being addressed.

We thank you for your overall positive evaluation and recommendations. We have made substantial changes in the manuscript in the light of these comments and hope that these revisions have addressed all the concerns.

**Minor Comments:**

(1) Validation of the adjustments: Three types of adjustments including effective day length, drying factor, and overwintering were involved in the products, and it seems that such adjustments were empirical. For example, the settings of carry-over fraction and effectiveness of winter precipitation for overwintering seem subjective. Since the results (e.g., Fig., 3-5) showed considerable differences for the adjustments relative to the original ones, how to guarantee that the adjustments were more effective for deducing fire risks? If the adjusted ones were not more effective on reflecting fire risks, why people should use them instead of the original ones?

Thank you for this comment. The original algorithm was designed for Canada, providing an empirical assessment of the fire risk in this region. Though, several components of the original algorithm were not appropriate for other world regions. That is why these adjustments are meant to extend this original algorithm to other world regions. For instance, you mention Figure 3, showing differences brought by the adjustments on the effective day length. In the original algorithm, all grid points receive the same day length, whether they are in Canada or not. These adjustments aim at correcting, providing the effective day length adequate for each grid point. Also, the Canadian Forestry Service that designed the original algorithm recommends using overwintering. As such, the adjusted algorithm is more effective at calculating fire risks in other world regions, and people should use the adjusted ones instead of the original ones.

We understand the reviewer's concern on "why people should use them instead of the original ones". We have now clarified the text in Abstract on this aspect as follows:

Lines 12-16:

*"Therefore, in this study we calculate and provide for the first time the Canadian Fire Weather Index (FWI) with all available simulations of the 6th phase of the Coupled Model Intercomparison Project (CMIP6). Furthermore, we expand its regional applicability by combining improvements on the original algorithm for the FWI from several packages."*

(2) Average versus minimum relative humidity: I understand that there are larger ensembles for FWI using average relative humidity, but it seems not clear whether such FWI based on average relative humidity achieve reasonable performance on reflecting fire risks relative to that based on minimum relative humidity. Since Fig.6 showed noticeable differences between these two FWIs and the annual indicators such as "fwixd" were based on the exact

FWI values, it is reasonable for the potential users to know whether they used FWIs were reliable or not.

We thank the reviewer for pointing this out. The Anonymous Referee #2 shared concerns on this aspect as well. For these reasons, the focus in this manuscript has shifted from the average relative humidity to the minimum relative humidity. The sensitivity analysis and the description of results are now based on the FWI calculated with minimum relative humidity, and the FWI using average relative humidity is now only featured as another possibility for users seeking more members, and described in the sensitivity analysis to this variable. Both datasets remain unchanged in the archive.

(3) Usage notes: the possible paths for data users deserve more explanation or discussion. It is important for the users to clearly know the usage of the data and what kinds of highly urgent scientific questions can be answered with the produced data. For example, why the listed opportunities in line 410-420 are important? what kinds of scientific questions remained in the fire weather studies but could be answered with the produced data?

We are grateful for this comment. We have developed this paragraph to explain further how these opportunities are indeed important research questions, and how this new dataset may be of use.

Lines 437-483:

*"The provided data is produced by the Institute of Atmospheric and Climate Science Institute of ETH Zurich. It is an open source and entirely free dataset. To illustrate possible paths for data users, we indicate in the following list some of the many opportunities where this dataset could be used. Some may rather be considered as research questions while some other points may be of interest for societal issues regarding fires.*

*As detailed in Section 2, we highlight that CMIP6 data may come with biases, while observations provide more realistic inputs and information for fire related studies. Though, observations have lower temporal and spatial availability and cover only the historical period. Thus model-based data facilitates large scale analysis.*

- *Comparison of FWI results with observations to evaluate the biases in the models. Compared to observations, some models show biases in their outputs. How does that affect the calculation of a compound product like the FWI? The FWI can be calculated using data based either on models or on observations (e.g. (Vitolo et al., 2019)). One may use the dataset provided here to evaluate the discrepancies and eventually how it affects future projections in fire weather. A first work in this direction has been produced with 16 ESMs and 1 ensemble member over the historical period (Gallo et al., 2022).*
- *Discrepancies in FWI across ESMs projections. The ESMs show different regional evolutions in some variables, though the effect of these discrepancies on the FWI remains unclear. One may investigate how much do projections in fire weather depend on the ESM by using the provided dataset and investigate reasons for the (dis)agreements.*
- *Dependencies of the FWI to ensemble members. The former path could be extended to the ensemble members. An uncertainty in the projections of the FWI arises from the initial conditions as well. The provided dataset may be used to assess this uncertainty and eventually the natural variability in FWI.*

- *Dependencies of the FWI to scenarios. Another dimension of projections in the FWI is the choice of the scenario. Under low warming scenarios, the Earth system gets more time to stabilize, allowing for different regimes, e.g. in the water cycle. It may help to investigate the response of the fire regimes across different scenarios. For example, the differences between low warming or high warming scenarios or even overshoot scenarios can be investigated using the provided dataset.*
- *It can be used to understand the effects of humidity regimes on fire regimes: minimum relative humidity and average relative humidity have different dynamics, and it is still unclear how they may affect the dynamics of fire weather in current and future climate. The provided dataset may help in assessing these regimes and their differences.*
- *Comparison of climatology of FWI in preindustrial, current, and future climate. Figure 8 of this manuscript gives a brief overview of this path. What should we expect from fire weather at different levels of climate change? Such a question would be of interest to inform society for the implications of climate change, and the provided dataset may help to answer it.*
- *Relationship of fire weather to modeled burned area. There is literature showing the correlation between FWI and burned area(Jones et al., 2022), in spite of other relevant factors such as fire ignition. One may use the provided dataset to check in the CMIP6 ensemble whether these relationships could be improved, and how they could be used, e.g. in impact models.*
- *Attribution studies of FWI to anthropogenic climate change under historical and future projections. Heatwaves, droughts and other extreme events have been attributed to climate change, but only limited studies have been able to attribute fires or mega-fires to climate change. The lack of relevant data explains this reduced number of attribution studies. Thanks to this provided dataset, attribution studies may use this data to assess changes in probabilities due to climate change. Though, the provided dataset does not provide runs under the scenario "hist-nat", the historical run with only natural forcings but not anthropogenic forcings. It remains possible to use this dataset by considering pre-industrial period and current period with their corresponding natural variability.*
- *FWI under CMIP5 and CMIP6. The FWI has been calculated for CMIP5 runs in (Abatzoglou et al., 2019), while the provided dataset calculates the FWI for the latest CMIP6 exercise. A comparison of both datasets would allow us to identify changes in fire weather between the ESMs. Coupled to their respective burned areas, one may disentangle the causes for differences in fires under ESMs between fire modules and fire weather of the models."*

(4) The title: this study focused on the Canadian fire weather index data, therefore many other fire weather indexes in Table A (line 420) were not involved. So I suggest to revise the title to highlight "Canadian fire weather index".

We thank the Anonymous Referee #1 for this insightful comment. This index is indeed the "Canadian Fire Weather Index" and should therefore be called the CFWI. Though, its widespread use has led it to be simply named Fire Weather Index and FWI, despite the other indexes. This is the case for Bedia et al., 2018 or Abatzoglou et al., 2019 for work on the FWI at a large geographical scale, or for applications of the FWI in specific regions (papers cited lines 200-202).

Thus, we decide to comply to the uses of this term in the literature and apologize to Anonymous Referee #1

---

## Author Comment (AC2)

**Response to Anonymous Referee 2 for the manuscript**

Fire weather index data under historical and SSP projections in CMIP6 from 1850 to 2100

Yann Quilcaille[1*], Fulden Batibeniz[1*], Andreia F. S. Ribeiro[1], Ryan S. Padrón[1], Sonia I. Seneviratne[1]

[1]Institute for Atmospheric and Climate Science, Department of Environmental Systems Science, ETH Zurich, Zurich, Switzerland

*These authors contributed equally to this work

*Correspondence to*: Yann Quilcaille and Fulden Batibeniz (yann.quilcaille@env.ethz.ch and fulden.batibeniz@env.ethz.ch)

We would like to thank the reviewers for their valuable comments. We have addressed all comments of the Anonymous Referee #1, Anonymous Referee #2 and Anonymous Referee #3 through appropriate changes and hope that the revised manuscript satisfies the Referees' concerns.

The Response to the Referees file provides complete documentation of the changes made in response to each comment. While this comprehensive explanation requires some repetition of material throughout the answer, our intention is that it helps to evaluate how each comment has been addressed.

Referees' comments are shown in black. The authors' response is shown in green text. The text quoted from the manuscript is shown between quotation marks in italics. Numbers of lines correspond to the version including tracked changes.

Summary of modifications:
- Modification of abstract and introductions
- Highlighted the novelty of the dataset in abstract and introduction
- Extensive changes to the Usages Notes
- To answer concerns on RHmean vs RHmin, the paper has been rewritten to feature the data produced using RHmin as main dataset and the data produced using RHmean as secondary dataset.
    - Description of data: updated section 2.1 and Figures 1 & A1 exchanged
    - Sensitivity analysis: updated section 3.1, 3.2 & 3.3; updated Figures 3-5 and A2-A4
    - Results: updated section 3.5; updated Figures 7, 8 and A6.
    - Data: nothing changed, both datasets were already provided.
- Minor revisions in the text

The manuscript titled "Fire weather index data under historical and SSP projections in CMIP6 from 1850 to 2100" presents a global dataset of FWI changes over the long term, using all CMIP6 simulations. The data could be used to evaluate the impacts of climate change on fire danger. Although I appreciate the significant effort the authors have made in processing the CMIP6 data, calculating the FWI, and estimating the model agreement in

different regions, I have some major reservations about the study in its current form, which are detailed below.

We would like to thank the reviewer for their overall positive evaluation and recommendations. We have made substantial changes in the manuscript in the light of these comments and hope that these revisions have addressed all the concerns.

**Major Comments:**

(1) I would argue that it is a convention to use the RHmin (rather than RHmean) to calculate the fire weather index, as per previous studies (Van Wagner, 1987; Vitolo et al., 2019; Abatzoglou et al., 2019). Replacing RHmin with RHmean resulted in large decreases in DMC, FFMC, and FWI by 30-35% (Fig. 6). Especially, using RHmean influence the monthly variations of DMC and FWI, which is expected as there are much larger differences in RHmin than RHmean between the wet and dry seasons. The author chose to use RHmean as "because daily minimum relative humidity is not provided for many CMIP6 runs, reducing the total number of runs from 1486 to 1321 (Line 303-304)". I suggest that 1321 CMIP6 runs with RHmin would be adequate for conducting FWI predictions.

We thank the reviewer for providing their insightful feedback. We understand the concern regarding the use of RHmean and we are aware of the differences between these variables. We have included both datasets for two reasons. The availability of a large amount of data can facilitate the use of AI applications. Besides, having both datasets would enable users to investigate the impact of relative humidity range on fire weather or how different changes in regimes of relative humidity may affect regimes in fire weather. If a user decides to work on the usual applications for FWI, the provided dataset with RHmin would allow it.
Furthermore, we would like to address the reviewer's concern about the RHmin by using RHmin instead of RHmean:
  - The text of Section 2.1 has been updated, showing that the FWI data calculated with RHmin is now the main dataset provided, and the FWI data calculated with RHmean becomes the second dataset. Figures 1 and A1 are exchanged.
  - The text of Sections 3.1, 3.2 and 3.3 have been updated to use RHmin for the sensitivity analysis on the adjustments. The Figures 3, 4 and 5 and their counterpart on January 1st in the Appendix A2, A3 and A4 have been updated as well. We acknowledge that the figures look alike, but this is due to similar effects from adjustments on results based on RHmin and RHmean. Though, their levels are all shifted, especially during local fire seasons.
  - The text of Section 3.5 has been updated, now using results based on RHmin. Figures 7, 8 and A5 have also been updated.

(2) The manuscript needs to highlight the novel aspects of the FWI dataset, especially compared to the one produced by Abatzoglou et al. (2019), which generally describes FWI in the same period (1860-2099). What new information could be obtained using the CMIP6 ensembles?

We appreciate your comment. Indeed, we did not sufficiently highlight the novelty of our study in the Introduction section. This study includes the first FWI index produced using the CMIP6 dataset.

Using a database based on the CMIP6 ensembles has several interests. First, this is the latest modeling exercise, thus accounting for the efforts in developing CMIP5 ESMs to CMIP6 ESMs. Then, not only the models have changed, but also the projections. The SSP-RCP framework is meant to map the mitigation and adaptation space, thus of interest for research questions related to fire weather. Finally, the CMIP6 exercise had more Tier 1 and Tier 2 variables, leading to a greater number of runs and variables to better understand processes related to fires.

The remarkable description of the FWI under the same period by Abatzoglou et al., (2019) could allow a comparison of CMIP5 and CMIP6 modeling efforts. Though, our manuscript differs slightly, in that we aim at providing this database to the community, instead of describing the FWI. Thus, a comparison of CMIP5 to CMIP6 runs would need to gain access to results from Abatzoglou et al., (2019).

We have now added new text that describes the novelty of our study in the abstract and introduction. The revised text states;

Lines 12-16:
*"Therefore, in this study we calculate and provide for the first time the Canadian Fire Weather Index (FWI) with all available simulations of the 6th phase of the Coupled Model Intercomparison Project (CMIP6). Furthermore, we expand its regional applicability by combining improvements on the original algorithm for the FWI from several packages."*

Lines 78-84:
*"Here, we present a new dataset of FWI, based on climate data from the 6th phase of the Coupled Model Intercomparison Project (CMIP6) and using an improved algorithm. We build upon the work of (Abatzoglou et al., 2019) for the previous generation of CMIP models. The novelty of this work comes from (1) the expanded regional applicability thanks to improvements on the original algorithm, (2) using the latest CMIP data covering historical and shared socioeconomic pathways (SSPs), from 1850 to 2100, and (3) providing the whole database to the users, thus enabling a large range of usages."*

(3) The manuscript requires further explanation of the methods, validation, interpretation of the results, and discussions of the data limitations. Specifically, the authors need to clarify the following:
(3.1) Why do we need to use the "day length" and "drying factor" adjustments? How do these adjustments play a role in different seasons? The authors need to provide equations to describe the adjustment explicitly.

We are grateful for this comment, reflected as well by Anonymous Referee #1. The original algorithm for the Canadian Fire Weather Index is based on empirical work in Canada. This work has made the FWI one of the most used indexes for fire weather. However, several aspects of this original code hinders its regional applicability. This is the case of the effective day length and the drying factor. In the original algorithm, all grid cells would receive the same value, whether they are in Canada or not. Hence, these adjustments aim at bringing appropriate values for the day length or the drying factor, depending on the latitude, the month or even the day of the year. This is why these adjustments are needed.

You are right in saying that these adjustments play different roles in different seasons. This is what we are illustrating in panels (g) to (l) of Figures 3 to 6.

Regarding equations, these values are originally not based on equations and are sets of values. That is why we decided to shortly describe all adjustments in Table 1, instead of providing hardly comprehensible lists of values to the readers.

(3.2) How is this FWI product compared with other FWI products (Vitolo et al., 2019; Abatzoglou et al., 2019)? A comprehensive evaluation is needed in the historical period to conduct future analyses.

We thank the reviewer for bringing the work of Vitolo et al., (2019) up. The Anonymous Referee #3 shared this opinion. It would be indeed very useful to compare the historical period of our results to the ERA5-Interim reanalysis dataset produced by Vitolo et al., 2019. However, our aim is to provide a dataset to enable many applications in our community, including evaluating its compatibility with FWI products based on reanalysis data or observations.

A study on this topic was actually submitted very shortly before this manuscript, which came to our attention only now. It examines how well 16 GCMs from the CMIP6 simulate fire weather indicators from the Canadian Forest Fire Weather Index System between 1979-2014 period (Gallo et al., submitted, https://doi.org/10.5194/gmd-2022-223). This work finds that, globally, the ensemble mean represents the variability, magnitude, and spatial extent of fire weather indicators reasonably well, compared to the latest global fire reanalysis. However, the performance of each GCM varies by region and season. The authors have done this evaluation only over the historical period, one single ensemble member, 16 GCMs and without providing the database. In our case, we obtained a total of 1486 runs.

Reproducing this comparison with the full ensemble would duplicate their work, although with a much higher size. We kindly ask the reviewer to understand that this amount of work is not feasible in this manuscript. Therefore, we acknowledge that this different research question is indeed interesting and recommend this ambitious work for future users in the Usage Notes while highlighting Vitolo et al, 2019 as a product to do this comparison and Gallo et al., (submitted) as an example. The text stating the is as follows;

Lines 61-62:
*"Historical fire weather can be investigated with observations, remote sensing products or more spatially and temporally homogeneous reanalysis datasets (Vitolo et al., 2019)."*

Lines 443-449:
*"Comparison of FWI results with observations to evaluate the biases in the models. Compared to observations, some models show biases in their outputs. How does that affect the calculation of a compound product like the FWI? The FWI can be calculated using data based either on models or on observations (e.g. (Vitolo et al., 2019)). One may use the dataset provided here to evaluate the discrepancies and eventually how it affects future projections in fire weather. A first work in this direction has been produced with 16 ESMs and 1 ensemble member over the historical period (Gallo et al., 2022)."*

Regarding Abtzoglou et al., 2019, we would also like to compare the outputs under this paper and our work, yet the data was not provided in Abatzoglou et al., 2019. However, we acknowledge that one may envisage to do this work, equally ambitious, in the Usage Notes:

Lines 480-483:
*"FWI under CMIP5 and CMIP6. The FWI has been calculated for CMIP5 runs in (Abatzoglou et al., 2019), while the provided dataset calculates the FWI for the latest CMIP6 exercise. A comparison of both datasets would allow us to identify changes in fire weather between the ESMs. Coupled to their respective burned areas, one may disentangle the causes for differences in fires under ESMs between fire modules and fire weather of the models."*

(3.3) As we know, increases in FWI could not be translated to the burned area change directly. Then how could we interpret the increase in FWI? Please extend more discussions in section 3.5 than just providing the numbers. Also, why do we see less agreement in FWI changes in boreal Asia, boreal America, African tropical forests, and India?

We thank you for your useful comment. Although the relationships of FWI to burned areas do not show perfect correlations, there are some regions where the increase in FWI can actually be translated to burned area change directly. The papers mentioned in the Introduction propose different relationships, and we list here the figures for the map of their Pearson's coefficients:
  - Figure 3 of Bedia et al, 2015 (10.5194/esd-10-91-2019)
  - Figure 2 of Abatzoglou et al, 2018 (10.1111/gcb.14405)
  - Figure 1 of Grillakis et al, 2022 (10.1088/1748-9326/ac5fa1)
  - Figure 7 of Jones et al, 2022 (10.1029/2020RG000726)
The FWI represents only the components related to the fire weather, but there are indeed other factors at play, as detailed in the Introduction. Specificities to the region or even human factors or will affect dynamics of wildfire. Deducing the precise implications for burned area represents indeed a very interesting work, but that goes beyond the scope of this data paper, similarly to Vitolo et al., 2019. Nevertheless, we acknowledge in the Usages Notes that this work would be important:

Lines 468-472:
*"Relationship of fire weather to modelled burned area. There is literature showing the correlation between FWI and burned area(Jones et al., 2022), in spite of other relevant factors such as fire ignition. One may use the provided dataset to check in the CMIP6 ensemble whether these relationships could be improved, and how they could be used, e.g. in impact models."*

The regional (dis)agreement may be related to discrepancies in CMIP6 models, e.g. different trends in precipitation or different seasonalities. Analyzing the reasons for these lower agreements represents a challenge in itself, and we have to accept that this can not be part of the scope of this data paper. Though, we gladly acknowledge that this path cannot be explored in the Usage Notes.

Lines 450-452:
*"Discrepancies in FWI across ESMs projections. The ESMs show different regional evolutions in some variables, though the effect of these discrepancies on the FWI remains unclear. One may*

*investigate how much do projections in fire weather depend on the ESM by using the provided dataset and investigate reasons for the (dis)agreements."*

We acknowledge that this is the fourth time that we explain that your comment would be interesting to treat, but falls within the Usages Notes. We apologize that it might seem like a lack of will, but this is not. Actually, we would be truly happy to do these works, but all these questions are research questions worth a publication each. This manuscript being submitted to a data journal, we aim at bringing data, while striving for its best quality possible. As far as we know, this dataset is the only one, based on model runs, with such an ensemble and open-source. It limits drastically the potential for comparison. Besides, such a sensitivity analysis has never been performed before. The objective of this data paper is not to exhaust potential applications of this dataset before users can access it, but to explain the process to create this data and the ensuing applications. We plan on using this dataset in the future, but the whole community would benefit by having users employing this data for their research.

(4) Clarify the data limitations for readers who will use this data for analyses.

We thank the reviewer for this suggestion. In the light of this comment we added a new text mentioning the uncertainties and limitations the CMIP6 can produce in the Usage Notes sections. The new text states:

Lines 111-117:
*"We highlight that using CMIP6 data comes with limitations. Although this is the result of a large community effort (Tebaldi et al., 2021), there may be some biases and discrepancies in these inputs (Wilcox and Donner, 2007; Rossow et al., 2013; Pfahl et al., 2017; McKitrick and Christy, 2020). Analysis of these biases have been performed for temperatures in (Fan et al., 2020), regional precipitations (Rivera and Arnould, 2020; Agel and Barlow, 2020; Ajibola et al., 2020), relative humidity (Douville et al., 2022) and wind (Shen et al., 2022). A bias-corrected version of CMIP6 data may be used as inputs, but existing datasets do not provide the necessary variables for the computation of the FWI (Carvalho et al., 2021; Xu et al., 2021), nor the full ensemble that we use here."*

Lines 441-443:
*"As detailed in Section 2, we highlight that CMIP6 data may come with biases, while observations provide more realistic inputs and information for fire related studies. Though, observations have lower temporal and spatial availability and cover only the historical period. Thus model-based data facilitates large scale analysis."*

**Comments on figures:**
**Figure 3:**
(1) I suggest that the authors show the changes in the fire season of the Southern Land which has a greater difference, for example, January 1st instead of July 1st. They could exchange Figure A2 and Figure 3.

We thank the reviewer for this suggestion. Showing the maps on January 1st (current Figure A2) instead of July 1st (current Figure 3) would indeed mean showing the maps during the fire season in the Southern hemisphere. However, it would also not be during the fire season

in the Northern hemisphere. This is then a matter of a choice, whether we show the maps during the Northern Hemisphere's fire season (original) or during the Southern Hemisphere's fire season (suggested). We decide to keep the maps for July in the main text and the maps for January in the Appendix, thus both fire seasons.
Our choice is motivated by overwintering, which concerns mostly Northern land. On the 1st January (Figure A4), there are barely differences to see, but much more interesting on the 1st July (Figure 5). To conserve consistency in the manuscript and improve its understandability by readers, we conserve the same date for these maps.

(2) Does the shaded area in Figs. (g-l) show ±1 standard deviation for historical only or both historical and SSP585. The shaded areas get so overlapped with each other and confuse me what can be learned from the figure. And there is no interpretation of this standard deviation range.

Thank you very much for this comment. The shaded areas do show both +/- 1 standard deviation for historical and SSP5-8.5. They overlap because there are no strong changes in this range. There are still some differences (e.g. in the upper range of (g), (i) and (k) of Figure 3). We decided to focus on the mean response to avoid losing the readers in even more numbers than there are already.

**Figure 4:** Again. I suggest exchanging Figure 4 and Figure A3 because the regions (Southern Land) showing large differences are in the wet season on July 1st. Therefore, we need to focus on fire season changes.

Thank you very much for this recommendation. Our answer to this comment is the same with the answer on Figure (3) (1) comment. Representing on July 1st or January 1st is a choice of representing the fire season in the Northern Land or the Southern Land. We chose the former, because of the figure of overwintering and to preserve consistency with the other figures.

**Figures 7 & 8:**
(1) Can you explain why you use 1851-1900 as a reference year? Is it better to use more recent years (e.g., 2000-2020) when the observations of most fire regimes are available?

Thanks for your insights. We selected this period to define a period without a climate change signal.  This way of analyzing the projections for different global warming levels (GWLs, +1°C, +1.5°C, +2°C and +3°C) with respect to pre-industrial conditions is consistent with the 6th Assessment Report of the Intergovernmental Panel on Climate Change (IPCC AR6) framework. Global warming levels are calculated over 20-years periods centered on the year of exceedance of the GWL (Text A.1). This method responds to their different climate sensitivity and internal variability. Global warming levels are also calculated according to this period to avoid scenario dependent uncertainty.

(2) The authors need to clarify how they calculate the number of days with extreme fire weather, the length of fire season, and the seasonal average of the FWI at different GWLs. For example, are you using the number of days above 95-th percentile of the FWI over 1851-1900 to calculate fwixd at different GWLs?

We thank you for this comment. The methods are already entirely described. The method for the calculation of the annual indicators are provided in Section 2.3, lines 177-183. The method for the calculation of each annual indicator at different GWLs is entirely provided in Text A.1, lines 489-530.

To answer your specific example, 1850-1900 is used to calculate the local 95-th percentile of the FWI (1851 instead of 1850 would not significantly affect the value of this threshold). Then, the number of days above this threshold are counted every year, providing fwixd. When calculating fwixd in a specific run at a specific GWL, we select the maps over a 20-years window around the year of exceedance of the GWL (all details line 493-506) and average these maps to obtain fwixd at this GWL for this run. Thus, in this case, the only way that we are using 1851-1900 is as a reference period for the change in global mean surface temperature. In the special case where we are calculating fwixd during 1851-1900, this is not a calculation at GWL, but simply the maps of fwixd averaged over this period. One may expect about 5% of 365.25 days, thus about 18.26 days on average. The fluctuations observed in Figures 7 and 8 are due to 1850, being included in the definition of the threshold and not in the calculation of this reference period, thus changing the results by about 1%.

**Minor Comments:**
(1) Line 130: what are the benefits of using this day length adjustment?

This day length adjustment aims at correcting the lack of dependency of the day length to the latitude and the day of the year. In the original algorithm, a day in January in Canada would last as long as a day in Brazil or New Zealand, and would not change over the whole month. This is why several packages proposed their versions, so we did this sensitivity analysis and showed these effects.

(2) Line 133: it would be helpful to explain how the day length parameter varies across different seasons and whether it only affects fire season.

We are grateful for this comment, we had internally similar questions on how to represent these effects.
The effective day length is periodic over the year, but not constant with latitude and different depending on the adjustments. We decided to summarize these adjustments in Table 1 without going too much into the details of what every package is doing.
Their effect on the fire season is shown in Figure 3, through panels (g) to (l).

(3) Line 137: I am curious about the reason for considering potential ET. Can you provide an additional explanation?

Thank you for your comment. The original algorithm from the Canadian Forestry Service uses this parameter to infer the potential ET. This is why we mention it here. Modeling the potential ET may be done through different means, but this is the empirical relationship chosen by the creators of the original algorithm.

(4) Line 205: Please refrain from using "correct" here –it is unclear whether adding the adjustments would improve the FWI prediction or not

We are grateful for this comment. It is true that at this time of the manuscript, we have not shown yet how adjustments to the original day length inadequate for most latitudes are changing the results. We have edited the text as follows:

Line 211:
*"The adjustments mostly change values in the Southern Hemisphere, where the effective day lengths were not prepared in the original calibration."*

(5) Line 205-210: In Fig. 3, this is a clear seasonable pattern in DMC in the Northern land and Tropical land, but not in the Southern land. Can you explain why?

We thank you for your attentive comment. We acknowledge a lower seasonality for DMC in Southern land. A potential explanation would be that through the DMC algorithm and its timescales, the combined seasonality in precipitation and relative humidity in these regions may be different, especially in the Amazonia, Central Africa and Indonesia. Though, such an intuition would need to be confirmed in future studies.

(6) Line 214: Why is there a decrease in the range of FWI/DMC?

Using the first adjusted version instead of the original algorithm affects the average DMC and FWI. This effect is lower at the end of SSP5-8.5 compared to the end of the historical period. It may be due to a stronger drying regime in SSP5-8.5 compared to the historical, causing these adjustments affecting the drying not to matter as much. Though, this is only an intuition and would need to back up with research on this specific question. We have edited the text to suggest this explanation:

Lines 234-235:
*"This reduction might be due to the stronger drying regime at the end of ssp585 compared to the historical period, causing this adjustment affecting drying not to matter as much."*

(7) Line 248-249: Could you rephrase the sentence where you mention "one month before the observed extreme in the differences in DC"?

Thank you for this comment. You are right in saying that this sentence should be rephrased, we did so by avoiding the term extreme:

Lines 269-270:
*"The adjustment to the drying factor in the Southern land is at its peak in July and at its minimum in January, one month before the observed maxima in the differences in DC."*

(8) Line 253: "However, the FWI presents higher sensitivities to changes in FFMC than to DMC, and even more to DC": This sentence is confusing: not clear if the sensitivity of FWI to DC is higher or lower than FFMC. I think Dowdy et al. show FFMC > DMC > DC. Please rephrase here.

You are right in pointing this out. This is indeed what Dowdy et al., 2010 shows. We have rephrased here as follows:

Lines 274-275:
*"However, sensitivities of FWI are by increasing order to FFMC, then DMC and finally DC (Dowdy et al., 2010)."*

(9) Line 255: Please avoid using the word "correcting"

Thank you for reminding us to refrain using this word. We have rewritten this sentence, and went through the text for its other uses:

Lines 163-164:
*"The adjustment for overwintering uses the value of DC at the end of the fire season and the precipitation up to the start of the fire season, as defined in (McElhinny et al., 2020)."*

Lines 227-228:
*"Tropical and Northern land are less affected because the magnitude of the adjustment to the effective day length is smaller (Fig. 3 g-j)."*

Lines 276-278:
*"Even though this adjustment has a relatively low impact, for latitudes below 20°S, these adjustments help in adapting the climate effects on the most compact organic layers, which is of interest to reproduce seasonal cycles and long term effects of climate change (Van Wagner, 1987)."*

Lines 316-317:
*"The added value of overwintering is to balance the overestimation of spring moisture content if interrupting calculation of the FWI, or the underestimation of spring moisture content in uninterrupted calculation of the FWI."*

Lines 319-320:
*"Overwintering reduces the FWI by up to -18% during January-February and brings an important adjustment to DC."*

Line 322:
*"We consider that overwintering is necessary when adjusting this effect in full time series."*

(10) Line 343-344: "It concerns the length of the fire season, the annual maxima, and the seasonal average of the FWI, but not the number of days with extreme fire weather that continue to show an increasing trend in these regions". I still see an increase in the number of days with extreme fire weather (second row of Figure 7). Is my understanding incorrect?

We are grateful for your careful reading of this figure. In these regions, values on average are actually close to zero, but remain positive. The colorbar is adapted to each annual indicator, due to their respectives ranges. This is why the green on this row indicates low but positive values, but negative values for the others.

---

## Author Comment (AC3)

**Response to Anonymous Referee 3 for the manuscript**

Fire weather index data under historical and SSP projections in CMIP6 from 1850 to 2100

Yann Quilcaille[1*], Fulden Batibeniz[1*], Andreia F. S. Ribeiro[1], Ryan S. Padrón[1], Sonia I. Seneviratne[1]

[1]Institute for Atmospheric and Climate Science, Department of Environmental Systems Science, ETH Zurich, Zurich, Switzerland

*These authors contributed equally to this work

*Correspondence to*: Yann Quilcaille and Fulden Batibeniz (yann.quilcaille@env.ethz.ch and fulden.batibeniz@env.ethz.ch)

We would like to thank the reviewers for their valuable comments. We have addressed all comments of the Anonymous Referee #1, Anonymous Referee #2 and Anonymous Referee #3 through appropriate changes and hope that the revised manuscript satisfies the Referees' concerns.

The Response to the Referees file provides complete documentation of the changes made in response to each comment. While this comprehensive explanation requires some repetition of material throughout the answer, our intention is that it helps to evaluate how each comment has been addressed.

Referees' comments are shown in black. The authors' response is shown in green text. The text quoted from the manuscript is shown between quotation marks in italics. Numbers of lines correspond to the version including tracked changes.

Summary of modifications:
- Modification of abstract and introductions
- Extensive changes to the Usages Notes
- Highlighted the novelty of the dataset in abstract and introduction
- To answer concerns on RHmean vs RHmin, the paper has been rewritten to feature the data produced using RHmin as main dataset and the data produced using RHmean as secondary dataset.
    - Description of data: updated section 2.1 and Figures 1 & A1 exchanged
    - Sensitivity analysis: updated section 3.1, 3.2 & 3.3; updated Figures 3-5 and A2-A4
    - Results: updated section 3.5; updated Figures 7, 8 and A6.
    - Data: nothing changed, both datasets were already provided.
- Minor revisions in the text

The authors calculated the Canadian Fire Weather Index (FWI) with all available simulations of the CMIP6. A sensitivity analysis of the default versus the improved version shows significant

differences in final FWI. The authors recommended the one with average relative humidity for studies requiring large ensembles and the one with minimum relative humidity for studies focused on actual FWI values. They further found that at a global warming level of 3 degC, the mean fire weather would on average double in duration and intensity, while associated 1-in-10-year events would triple in duration and increase by half in intensity. The dataset and results are interesting and would be very helpful for the community. However, I do have a few concerns and questions for the authors to address.

We thank you for your overall positive evaluation and recommendations. We have made necessary changes in the manuscript in the light of these comments and hope that these revisions have addressed all the concerns.

**Major comments**

(1) My major concern is that all the datasets and results are purely model simulations without any evaluations of the model quantities used for FWI calculation or the FWI directly for historical periods. Without this important piece, it is difficult to convince readers how reliable and accurate this FWI dataset would be. I suggest at least doing some evaluations for those major model meteorological quantities used for FWI calculations and/or FWI directly (e.g., USGS FWI: https://www.usgs.gov/fire-danger-forecast/wildland-fire-potential-index-wfpi).

We are grateful for your comment. Our work is indeed purely based on model simulations. The objective of this paper is to provide a database to the community. Analyzing the whole CMIP6 database is a much bigger task, the work of several publications. We kindly ask the reviewer to understand that this is not feasible in such a data paper. Following your recommendations, we now include references for the reader for further information on this topic.

Lines 111-117:
*"We highlight that using CMIP6 data comes with limitations. Although this is the result of a large community effort (Tebaldi et al., 2021), there may be some biases and discrepancies in these inputs (Wilcox and Donner, 2007; Rossow et al., 2013; Pfahl et al., 2017; McKitrick and Christy, 2020). Analysis of these biases have been performed for temperatures in (Fan et al., 2020), regional precipitations (Rivera and Arnould, 2020; Agel and Barlow, 2020; Ajibola et al., 2020), relative humidity (Douville et al., 2022) and wind (Shen et al., 2022). A bias-corrected version of CMIP6 data may be used as inputs, but existing datasets do not provide the necessary variables for the computation of the FWI (Carvalho et al., 2021; Xu et al., 2021), nor the full ensemble that we use here."*

Comparing the FWI directly represents indeed less work, because it is only one dataset, by implicitly simplifying by not counting DC, DMC, FFMC, ISI and BUI). The Anonymous Referee #2 shared your opinion about a need for comparison to existing FWI products and suggested FWI products. It would be indeed very useful to compare the historical period of our results to observations-based FWI. The dataset that you kindly suggest (USGS FWI) is actually for another index, the Wildfire Fire Potential Index (WFPI), not the FWI. The data differs, the algorithm differs, the region is only for the US region and the final quantities have different

"units" and scales. Comparisons of the different fire indices exist and are cited in this manuscript.

However, a FWI data product based on the ERA5-Interim reanalysis dataset has been produced by Vitolo et al., 2019, which was brought to our attention by the Anonymous Referee #2. Yet, comparing a model-based FWI dataset to an observation-based FWI dataset would still remain an endeavor in our case due to the sheer size of the database.

A study on this comparison was actually submitted very shortly before this manuscript, which came to our attention only now. It examines how well 16 GCMs from the CMIP6 simulate fire weather indicators from the Canadian Forest Fire Weather Index System between 1979-2014 period (Gallo et al., submitted, https://doi.org/10.5194/gmd-2022-223). This work finds that, globally, the ensemble mean represents the variability, magnitude, and spatial extent of fire weather indicators reasonably well, compared to the latest global fire reanalysis. However, the performance of each GCM varies by region and season. The authors have done this evaluation only over the historical period, one single ensemble member, 16 GCMs and without providing the database. In our case, we obtained a total of 1486 runs.

Reproducing this comparison with the full ensemble would duplicate their work, although with a much higher size. We kindly ask the reviewer to understand that this amount of work is not feasible in this manuscript. Therefore, we acknowledge that this different research question is indeed interesting and recommend this ambitious work for future users in the Usage Notes while highlighting Vitolo et al, 2019 as a product to do this comparison and Gallo et al., (submitted) as an example. The text stating the is as follows;

Lines 61-62:
*"Historical fire weather can be investigated with observations, remote sensing products or more spatially and temporally homogeneous reanalysis datasets (Vitolo et al., 2019)."*

Lines 443-449:
*"Comparison of FWI results with observations to evaluate the biases in the models. Compared to observations, some models show biases in their outputs. How does that affect the calculation of a compound product like the FWI? The FWI can be calculated using data based either on models or on observations (e.g. (Vitolo et al., 2019)). One may use the dataset provided here to evaluate the discrepancies and eventually how it affects future projections in fire weather. A first work in this direction has been produced with 16 ESMs and 1 ensemble member over the historical period (Gallo et al., 2022)."*

A comparison to FWI from CMIP5 may have been produced thanks to Abtzoglou et al., 2019, although the data was not provided in Abatzoglou et al., 2019. However, we acknowledge that one may envisage to do this work, equally ambitious, in the Usage Notes:

Lines 480-483:
*"FWI under CMIP5 and CMIP6. The FWI has been calculated for CMIP5 runs in (Abatzoglou et al., 2019), while the provided dataset calculates the FWI for the latest CMIP6 exercise. A comparison of both*

*datasets would allow us to identify changes in fire weather between the ESMs. Coupled to their respective burned areas, one may disentangle the causes for differences in fires under ESMs between fire modules and fire weather of the models."*

(2) It is also not very clear to me what the key novelty of this dataset is. Is the algorithm relatively new compared to the previous methods? Is this the first global FWI dataset? Also, there needs to be some discussions in the introduction about the existing global/regional FWI or fire index datasets (if any).

We thank the reviewer for this comment. Indeed, we did not sufficiently highlight the novelty of our study in the Introduction section. There are two main points:
1. This study makes available for the first time the FWI index produced using the CMIP6 dataset, enabling many usages. Using a database based on the CMIP6 ensembles has several interests. First, this is the latest modeling exercise, thus accounting for the efforts in developing CMIP5 ESMs to CMIP6 ESMs. Then, not only the models have changed, but also the projections. The SSP-RCP framework is meant to map the mitigation and adaptation space, thus of interest for research questions related to fire weather. Finally, the CMIP6 exercise had more Tier 1 and Tier 2 variables, leading to a greater number of runs and variables to better understand processes related to fires.
2. Besides, the algorithm used is new in that it merges the improvements from different algorithms.

We acknowledge that we should have emphasized the novelty of this dataset. Following your recommendations, we are now writing:
Lines 12-16:
*"Therefore, in this study we calculate and provide for the first time the Canadian Fire Weather Index (FWI) with all available simulations of the 6th phase of the Coupled Model Intercomparison Project (CMIP6). Furthermore, we expand its regional applicability by combining improvements on the original algorithm for the FWI from several packages."*

Lines 78-84:
*"Here, we present a new dataset of FWI, based on climate data from the 6th phase of the Coupled Model Intercomparison Project (CMIP6) and using an improved algorithm. We build upon the work of (Abatzoglou et al., 2019) for the previous generation of CMIP models. The novelty of this work comes from (1) the expanded regional applicability thanks to improvements on the original algorithm, (2) using the latest CMIP data covering historical and shared socioeconomic pathways (SSPs), from 1850 to 2100, and (3) providing the whole database to the users, thus enabling a large range of usages."*

Regarding your recommendation on the existing datasets, we have added references to Vitolo et al., (2019), which provides an excellent and available dataset for comparison. We also mention the incoming comparison paper from Gallo et al., (submitted).

Lines 444-449:

"Comparison of FWI results with observations to evaluate the biases in the models. Compared to observations, some models show biases in their outputs. How does that affect the calculation of a compound product like the FWI? The FWI can be calculated using data based either on models or on observations (e.g. (Vitolo et al., 2019)). One may use the dataset provided here to evaluate the discrepancies and eventually how it affects future projections in fire weather. A first work in this direction has been produced with 16 ESMs and 1 ensemble member over the historical period (Gallo et al., 2022)."

**Minor comments:**

(1) Please clarify the spatial and temporal resolutions of the dataset in the abstract.

Thank you. We have added this information to the abstract.
Lines 25-27:
*"Ultimately, this new fire weather dataset provides a large ensemble of simulations to understand the potential impacts of climate change spanning a range of shared socioeconomic narratives with their radiative forcing trajectories over 1850-2100 at annual and 2.5° x 2.5° resolutions."*

(2) Section 2.1: please provide the spatial resolution for the CMIP6 model simulations.

Thank you for your comment. We understand that you may feel it is necessary to include a comprehensive list of information; however, as we are already sharing the data regridded to a common grid for the reader (2.5°X2.5° resolution), we refer to a study where they can find detailed information about the original grid resolution of each model (Tebaldi et al., 2021 https://doi.org/10.5194/esd-12-253-2021).

Line 109:
*"We highlight that using CMIP6 data comes with limitations. Although this is the result of a large community effort (Tebaldi et al., 2021)"*